# Cas12a2 elicits abortive infection through RNA-triggered destruction of dsDNA

Oleg Dmytrenko[1], Gina C. Neumann[2], Thomson Hallmark[3], Dylan J. Keiser[3], Valerie M. Crowley[3], Elena Vialetto[1], Ioannis Mougiakos[1], Katharina G. Wandera[1], Hannah Domgaard[3], Johannes Weber[1], Thomas Gaudin[1], Josie Metcalf[3], Benjamin N. Gray[2,5], Matthew B. Begemann[2✉], Ryan N. Jackson[3✉] & Chase L. Beisel[1,4✉]

Bacterial abortive-infection systems limit the spread of foreign invaders by shutting down or killing infected cells before the invaders can replicate[1,2]. Several RNA-targeting CRISPR–Cas systems (that is, types III and VI) cause abortive-infection phenotypes by activating indiscriminate nucleases[3–5]. However, a CRISPR-mediated abortive mechanism that leverages indiscriminate DNase activity of an RNA-guided single-effector nuclease has yet to be observed. Here we report that RNA targeting by the type V single-effector nuclease Cas12a2 drives abortive infection through non-specific cleavage of double-stranded DNA (dsDNA). After recognizing an RNA target with an activating protospacer-flanking sequence, Cas12a2 efficiently degrades single-stranded RNA (ssRNA), single-stranded DNA (ssDNA) and dsDNA. Within cells, the activation of Cas12a2 induces an SOS DNA-damage response and impairs growth, preventing the dissemination of the invader. Finally, we harnessed the collateral activity of Cas12a2 for direct RNA detection, demonstrating that Cas12a2 can be repurposed as an RNA-guided RNA-targeting tool. These findings expand the known defensive abilities of CRISPR–Cas systems and create additional opportunities for CRISPR technologies.

All domains of life use defence strategies that cause cells to enter dormancy or die to limit the spread of infectious agents[1]. In bacteria and archaea, this strategy is called abortive infection, and it is used by a vast variety of bacterial defence systems[1,2]. Recently, it was shown that CRISPR RNA (crRNA)-guided adaptive immune systems that target RNA cause abortive-infection phenotypes[3–6]. Type VI systems non-specifically degrade RNA, whereby the Cas13 single-effector nuclease acts as both a crRNA-guided effector and indiscriminate RNase[3,7,8]. In type III systems, target RNA binding triggers the production of cyclic oligoadenylate secondary messengers that in turn activate indiscriminate accessory RNases and ssDNases that can drive abortive infection[4,5,9–11]. Moreover, it has been proposed that abortive infection is mediated by indiscriminate dsDNases (such as NucC) activated through type III secondary messengers[12,13] or by indiscriminate ssDNase activity from type V Cas12a single-effector nucleases[14]. However, type III CRISPR-mediated dsDNase activity has yet to be examined in vivo, and the ssDNase activity of Cas12a was recently shown to not cause abortive infection[15].

Here we report that Cas12a2, a type V single-effector CRISPR-associated (Cas) nuclease, induces an abortive-infection phenotype when challenged with plasmids that are complementary to crRNA guides. Biochemical assays using recombinant protein revealed that Cas12a2 recognizes RNA targets, unleashing non-specific dsDNA-, ssDNA- and ssRNA-nuclease activities distinct from those of other single-subunit RNA-targeting (such as Cas13a) and dsDNA-targeting (such as Cas12a) Cas nucleases[8,16,17]. Furthermore, we show that the Cas12a2 non-specific nuclease activities damage bacterial DNA, triggering the SOS response and impairing cell growth. Collectively these results suggest that the dsDNase activity of Cas12a2 is instrumental in triggering the abortive-infection phenotype. As a proof-of-principle demonstration, we show that Cas12a2 can detect RNA at a sensitivity that is comparable to that of the RNA-targeting Cas13a nuclease at various temperatures.

## Cas12a2 induces abortive infection

Cas12a2 comprises a group of type V effector nucleases that are related to Cas12a[16], with Cas12a2 orthologues previously being classified as Cas12a variants[18]. Our analyses similarly place them in a monophyletic clade that shares the last common ancestor with Cas12a nucleases (Fig. 1a and Extended Data Fig. 1). Further analysis revealed that CRISPR–Cas12a2 and CRISPR–Cas12a systems feature CRISPR repeats with a conserved 3′ end, and the nucleases possess homologous RuvC endonuclease domains and a similar predicted secondary structure in the N termini (Fig. 1b and Supplementary Fig. 2). Despite the conserved RuvC-like domains and N termini, Cas12a2 is distinguished from Cas12a by the presence of a large domain of unknown function located in place of the Cas12a bridge helix as well as a zinc-finger domain in place of the

[1]Helmholtz Institute for RNA-based Infection Research, Helmholtz Centre for Infection Research, Würzburg, Germany. [2]Benson Hill, St Louis, MO, USA. [3]Department of Chemistry and Biochemistry, Utah State University, Logan, UT, USA. [4]Medical Faculty, University of Würzburg, Würzburg, Germany. [5]Present address: Syngenta, Research Triangle Park, NC, USA. ✉e-mail: mbegemann@bensonhill.com; ryan.jackson@usu.edu; chase.beisel@helmholtz-hiri.de

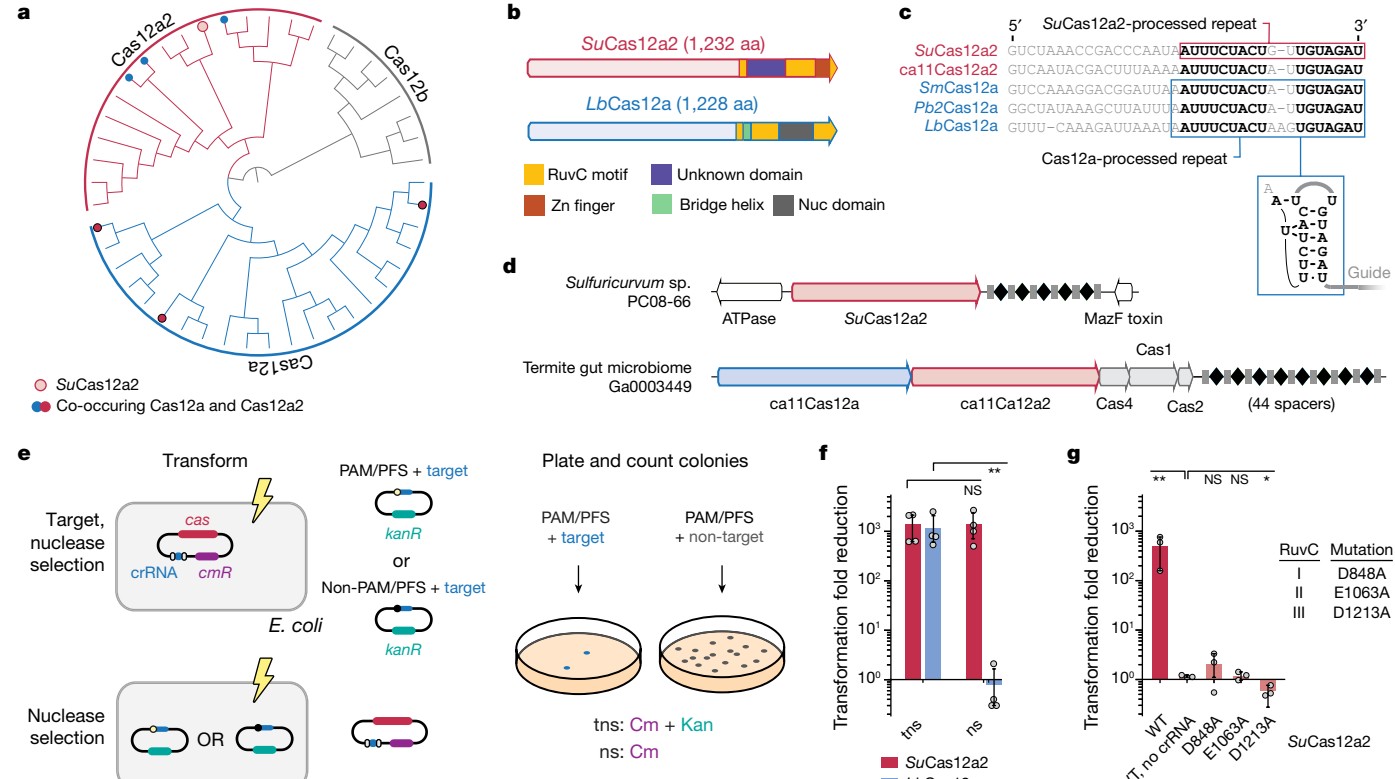

**Fig. 1 | Cas12a2 nucleases form a distinct clade within type V Cas12 nucleases. a**, Maximum-likelihood phylogeny of identified Cas12a2 nucleases with Cas12a and Cas12b nucleases. The detailed phylogeny is shown in Extended Data Fig. 1. Systems with co-occurring Cas12a2 and Cas12a are indicated by filled red and blue circles. *Su*Cas12a2 is indicated by an unfilled red circle. **b**, The domain architecture of *Su*Cas12a2 in comparison to *Lb*Cas12a. aa, amino acids. **c**, Aligned direct repeats associated with representative Cas12a2 and Cas12a nucleases. The bold nucleotides indicate conserved positions within the processed repeats for both nucleases. The predicted pseudoknot structure of the Cas12a repeat is shown below. The loop of the hairpin (grey) is variable. Pre-crRNA processing by *Su*Cas12a2 is shown in Extended Data Fig. 3. **d**, Gene organization of CRISPR–Cas systems within representative genomic loci encoding Cas12a2. Examples of systems encoding Cas12a2 as the sole Cas nuclease and those also encoding Cas12a are shown. **e**, Diagram of the traditional (top; target nuclease selection (tns)) and modified (bottom; nuclease selection (ns)) plasmid interference assay. Cm, chloramphenicol; Kan, kanamycin. **f**, The reduction in plasmid transformation for *Su*Cas12a2 and *Lb*Ca12a2 under target plasmid and nuclease plasmid selection. **g**, The reduction in plasmid transformation of *Su*Cas12a2 RuvC mutants under target plasmid and nuclease plasmid selection. For **f** and **g**, data are mean ± s.d. of at least three independent experiments started from separate colonies. *P* values were calculated using one-tailed Welch's *t*-tests; NS, $P > 0.05$; *$P < 0.05$, **$P < 0.005$.

Cas12a Nuc domain (Fig. 1b and Supplementary Fig. 2). Considering their original classification combined with our phylogenetic analyses as well as recent structural results[19], we named these distinct type V nucleases Cas12a2.

Notably, some CRISPR–Cas systems contain both *cas12a2* and *cas12a* genes in tandem next to a shared CRISPR array (Fig. 1c). From this observation and the conservation of CRISPR repeats from systems with either of the nucleases (Fig. 1d and Supplementary Fig. 3), we hypothesized that both proteins bind to and process similar crRNA guides. However, as the proteins diverge in other domains, we further hypothesized that Cas12a2 performs a defence function that is distinct from the dsDNA-targeting activity of Cas12a[16].

To test these hypotheses, we encoded the *cas12a2* gene from the sulfur-oxidizing epsilonproteobacterium *Sulfuricurvum* sp. PC08-66 (*Su*Cas12a2) along with a CRISPR array into an expression plasmid, which we introduced into *Escherichia coli* cells. We next performed a traditional plasmid interference assay that depletes cells by selecting for the plasmid containing the nuclease and crRNA as well as a target plasmid (Fig. 1e). This assay detects broad immune system activity but cannot distinguish between defence activities that only deplete the target from those that activate abortive-infection phenotypes. To test whether Cas12a2 uses an abortive-infection mechanism, we modified the assay by selecting only for the nuclease plasmid (Fig. 1e). Consistent with our hypothesis that Cas12a2 functions differently compared with Cas12a, Cas12a2 depleted cells in both the traditional (about 1,900-fold reduction) and modified (around 1,300-fold reduction) plasmid interference assays, whereas Cas12a from *Lachnospiraceae bacterium* (*Lb*Cas12a) depleted cells only in the traditional assay (Fig. 1f). Similar trends were observed with different targets cloned into the same plasmid location, different Cas12a2 homologues and when comparing *Su*Cas12a2 with the Cas12a homologue from *Prevotella bryantii B14* (*Pb*2Cas12a) (Extended Data Fig. 2a,b). Moreover, mutating predicted active residues within any of the three RuvC motifs in *Su*Cas12a2 impaired immune function (Fig. 1g). Collectively, these results indicate that Cas12a2 relies on a RuvC-nuclease domain and induces abortive infection through a mechanism that is distinct from that of Cas12a.

## Cas12a2 targets RNA and degrades dsDNA

CRISPR systems that cause abortive-infection phenotypes (such as types III and VI) rely on indiscriminate nucleases activated by RNA targeting[3–5]. To determine whether Cas12a2 uses a similar mechanism, we recombinantly expressed and purified *Su*Cas12a2 and tested its enzymatic activities in vitro (Fig. 2 and Supplementary Fig. 4). However, before examining the nucleic-acid-targeting activities, we needed to determine how Cas12a2 crRNAs are processed.

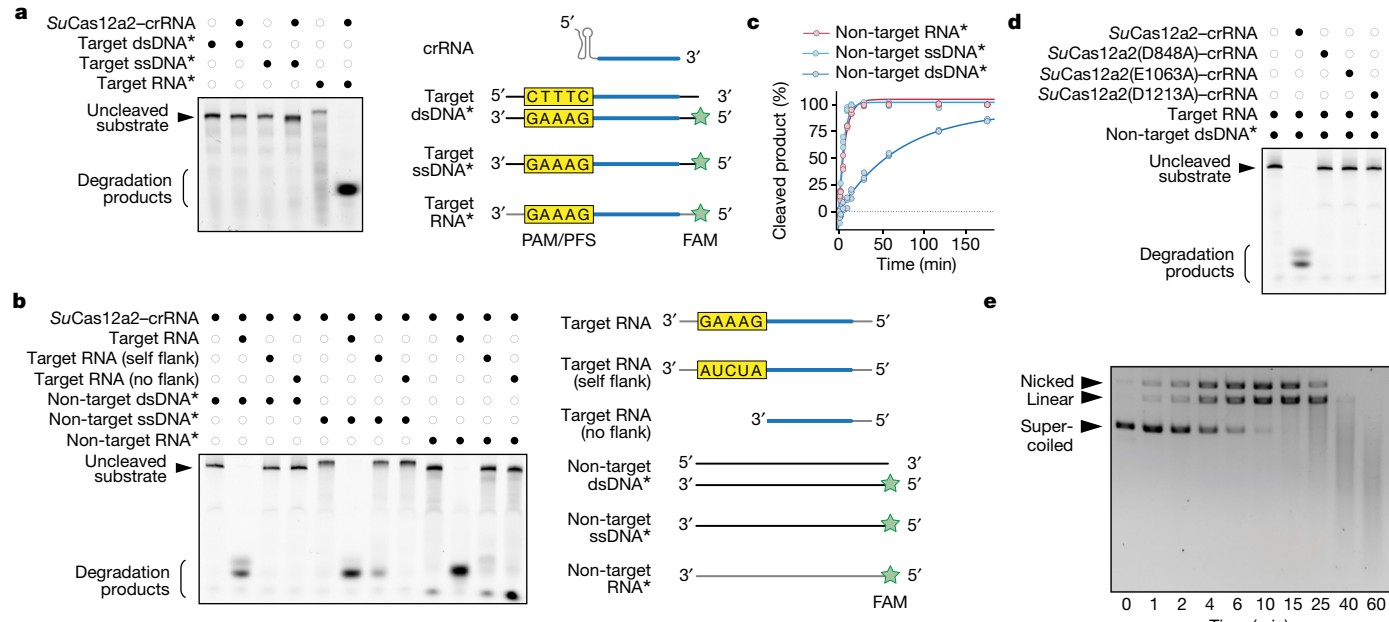

**Fig. 2 | RNA target recognition by *Su*Cas12a2 triggers degradation of ssRNA, ssDNA and dsDNA in vitro. a**, Direct targeting of different FAM-labelled nucleic-acid substrates by a purified *Su*Cas12a2–crRNA complex. **b**, Collateral cleavage of FAM-labelled non-target nucleic-acid substrates by the *Su*Cas12a2–crRNA complex with different target RNA substrates after 1 h. Target RNA, a non-self flanking sequence at the 3′ end; self flank, a flanking sequence mutated to the reverse complement of the crRNA repeat tag; no flank, only the reverse complement of the crRNA guide. For **a** and **b**, diagrams of target and non-target nucleic acids are shown on the right. **c**, Time-course analysis of RNA-triggered collateral cleavage of labelled non-target RNA, ssDNA or dsDNA.

Representative gel images are provided in Extended Data Fig. 4c. Note that dsDNA contains twice as much ssDNA substrate as the RNA and ssDNA, but the same concentration of labelled strands. **d**, The effect of mutating each of the three RuvC motifs on RNA-triggered collateral cleavage of dsDNA. **e**, Time-course analysis of RNA-triggered collateral cleavage of non-target plasmid DNA. Plasmid DNA was visualized using ethidium bromide. For **a**–**d**, the asterisks indicate a FAM-labelled substrate, and the diagrams on the right indicate the substrates. All of the results are representative of three independent experiments. Gel source data are provided in Supplementary Fig. 1.

The CRISPR repeats of the Cas12a and Cas12a2 systems are highly conserved at the 3′ end (Fig. 1d and Supplementary Fig. 3), and sequence alignments predict that Cas12a2 shares secondary structure in the region of the Cas12a pre-crRNA-processing active site[18,20] (Supplementary Fig. 5). Consistent with this prediction, RNA-sequencing analysis of pre-crRNAs processed by *Su*Cas12a2 in vitro revealed that processing occurs one nucleotide downstream of the position cleaved by Cas12a (Extended Data Fig. 3a,b). The 3′ end of the spacer also underwent trimming in vivo to form an approximately 24-nucleotide guide (Extended Data Fig. 3b,c), possibly through host ribonucleases as observed for Cas9 crRNAs[21]. Mutating basic amino acids (Lys784 and Arg785) located in the predicted RNA-processing active site abolished activity[22] (Extended Data Fig. 3d). Furthermore, plasmid interference assays revealed that Cas12a and Cas12a2 can interchange guides without impairing immunity (Extended Data Fig. 2c). Thus, the Cas12a2 nuclease processes its own crRNA guides like other type V effector nucleases[20,23] and can share crRNAs with Cas12a.

To determine the nucleic-acid target preference of crRNA-guided Cas12a2, complementary ssRNA, ssDNA and dsDNA substrates containing an A/T-rich flanking sequence (paralleling Cas12a substrates)[16,22] were fluorescently labelled with a FAM molecule and combined with crRNA-guided Cas12a2 (Fig. 2a). Similar to CRISPR–Cas systems that cause abortive infection—yet in contrast to the dsDNA-targeting Cas12a—Cas12a2 is activated only in the presence of complementary RNA targets. The potency of plasmid interference with *Su*Cas12a2 (Fig. 1f) was notable given the lack of a defined promoter upstream of the target in this construct. However, we attribute the interference to spurious transcription of the encoding plasmid for two reasons: introducing an upstream terminator significantly reduced plasmid interference in *E. coli* (Extended Data Fig. 2d,e), and an upstream promoter was required to detect collateral activity in a cell-free transcription–translation assay[24] (Extended Data Fig. 2f,g).

As other Cas abortive-infection mechanisms rely on collateral indiscriminate RNase activity, we examined whether specific RNA targeting by Cas12a2 induces indiscriminate nuclease activity. We found that *Su*Cas12a2 robustly and indiscriminately degraded FAM-labelled ssRNA, ssDNA and dsDNA substrates bearing no complementarity to the crRNA guide. By contrast, other Cas nucleases indiscriminately degrade only ssRNA (Cas13a)[8] or ssRNA and ssDNA (Cas12g)[25] after RNA targeting, or only ssDNA after dsDNA targeting (Cas12a)[14] (Fig. 2b and Extended Data Fig. 4). Of the three collateral substrates, ssRNA and ssDNA appear to be more efficiently cleaved than dsDNA by Cas12a2 (Fig. 2c and Extended Data Fig. 4b). However, this difference could be explained by the presence of twice as many DNA strands in dsDNA substrates compared with ssDNA substrates for the same amount of nuclease. Also, similar to Cas13a[8], complementary ssDNA and dsDNA do not activate any Cas12a2 non-specific nuclease activity (Extended Data Fig. 4a), and dsRNA is not a primary substrate of collateral cleavage (Extended Data Fig. 4c).

To examine whether Cas12a2 activity is reliant on detecting a 'non-self' signal adjacent to the target (called a protospacer-flanking sequence (PFS))[8], we performed in vitro cleavage assays in which target RNA sequences were flanked on the 3′ side with a 'self' sequence complementary to the crRNA repeat (5′-AUCUA-3′), the non-self PFS used in our in vivo assay (5′-GAAAG-3′) or a 'flankless' RNA complementary to the guide region of the crRNA, but containing no PFS (Fig. 2b). Notably, only the RNA target containing the non-self PFS activated collateral nuclease activity, demonstrating that specific nucleotides at the 3′ end of the RNA target must be present to activate the collateral activity of Cas12a2. Moreover, introducing disruptive mutations to any of the three RuvC motifs or conserved cysteine residues within the putative zinc-finger domain abolished all non-specific cleavage (Fig. 2d and Extended Data Fig. 4d), consistent with our in vivo plasmid interference results (Fig. 1g).

Our biochemical assays demonstrated that Cas12a2 could quickly remove a FAM label from linear dsDNA substrates, but it was unclear whether Cas12a2 degrades DNA lacking available 5′ or 3′ ends. We therefore challenged crRNA-guided Cas12a2 with an RNA target and a supercoiled pUC19 plasmid. Importantly, pUC19 does not contain any sequence complementary to the Cas12a2 crRNA guide. We observed that *Su*Cas12a2 rapidly nicked, linearized and degraded pUC19 DNA (Fig. 2e), but only in the presence of a cognate target and PFS and with an intact RuvC domain (Extended Data Fig. 4e). This rapid destruction of the supercoiled plasmid contrasts with the slow and incomplete linearization of plasmid DNA by Cas12a nucleases[26]. These data suggest a mechanism in which activated *Su*Cas12a2 is able to robustly hydrolyse the phosphodiester backbone of non-specific DNA regardless of whether it is supercoiled, nicked or linear. A comparison with Cas12a (dsDNA targeting with collateral ssDNase), Cas13a (ssRNA targeting with collateral ssRNase) and Cas13g (ssRNA targeting with collateral ssRNase and ssDNase) demonstrated that the RNA-targeting ssRNase, ssDNase and dsDNase are unique to *Su*Cas12a2 (Extended Data Fig. 4a). Collectively, these in vitro results reveal that crRNAs guide *Su*Cas12a2 to RNA targets, activating RuvC-dependent cleavage of ssRNA, ssDNA and dsDNA. These activities, in part or in total, may underlie the abortive-infection phenotype.

## Cas12a2 exhibits targeting flexibility

Although our in vitro data indicated an underlying mechanism for the Cas12a2 abortive-infection phenotype, we wanted to understand the targeting limitations of these distinct enzymes. In particular, we investigated the stringency of non-self PFS sequence recognition and penalties for mismatches between the crRNA and target. We therefore challenged *Su*Cas12a2 with a library of plasmids encoding all possible 1,024 flanking sequences at the 3′ end of the RNA target to the −5 position (Fig. 3a and Extended Data Fig. 5a,b). We found that *Su*Cas12a2 depleted approximately half of all of the sequences in the library, suggesting a PFS-recognition mechanism that is more stringent than that of Cas13 but still more promiscuous than those of most DNA-targeting systems[8,27]. The depleted sequences were generally A rich, consistent with a 5′-GAAAG-3′ PFS, but could not be fully captured by a single consensus motif (Fig. 3a and Extended Data Fig. 5c). We further validated individual depleted sequences, including representatives within five unique motifs recognized by *Su*Cas12a2 but not by *Pb*2Cas12a—a nuclease that is known for flexible PAM recognition[24] (Fig. 3b and Extended Data Fig. 5d). Consistent with its function as an RNA-targeting nuclease, the recognized sequences were broad but did not follow the expected profile if Cas12a2 is principally evaluating tag–anti-tag complementarity. These results further support a mechanism in which PFS recognition by *Su*Cas12a2 operates similar to type III systems that require recognition of a PFS or RNA PAM to activate[28–31] and distinct from the evaluation of tag–anti-tag complementarity used by RNA-targeting Cas13[8,32] and other type III CRISPR–Cas systems[33].

Most DNA- and RNA-targeting Cas nucleases have shown high sensitivity to mismatches within a seed region, in which a single mismatch between the crRNA guide and target disrupts binding[8,17,34,35]. Thus, to identify whether *Su*Cas12a2 relies on a seed region, we evaluated how *Su*Cas12a2 tolerates mismatches in our cell-based assay (Fig. 3c). Notably, *Su*Cas12a2 accommodated single and double mismatches across the target, with PFS-distal mutations exerting more adverse effects on plasmid targeting. To completely disrupt *Su*Cas12a2 targeting, four mismatches were needed throughout the guide (Fig. 3c) or up to 10 mismatches at the 3′ end of a 24-nucleotide guide (Extended Data Fig. 6a). The flexible PFS recognition and a tolerance for guide–target mismatches indicate that *Su*Cas12a2 exhibits promiscuous target recognition and appears to lack a canonical seed that is hypersensitive to guide–target mismatches[34,35]. However, the necessary pairing with the 3′ end of the guide is consistent with this end being pre-ordered in the structure of the crRNA–Cas12a2 binary complex[19] and possibly initiating base pairing with the target. Furthermore, the promiscuity further enabled *Su*Cas12a2 to recognize target mutations that disrupt targeting by Cas12a (Extended Data Fig. 6b).

Collectively, the distinct activities of Cas12a2 compared with Cas12a suggested that tandem systems possessing both nucleases (Fig. 1d) may act cooperatively to broaden the effectiveness against foreign viruses and plasmids. In particular, we hypothesized that the unique structural features of Cas12a2 might prevent the escape of viruses encoding anti-CRISPR proteins that block Cas12a function[36–38]. Consistent with this hypothesis, only one (AcrVA2.1) out of seven Cas12a anti-CRISPR proteins was able to impair Cas12a2 function, albeit only partially (Fig. 3d and Supplementary Fig. 6a,b). Notably, AcrVA5 also exhibited no inhibitory activity despite *Su*Cas12a2 possessing the conserved lysine residue in Cas12a that is chemically modified by this Acr to block PAM recognition[39] (Fig. 3d and Supplementary Fig. 6c). The limited ability of Cas12a Acr proteins to inhibit *Su*Cas12a2 further underscores the distinct properties of these nucleases and the ability of Cas12a and Cas12a2 to complement each other in immune defence.

## SOS response and dormancy by Cas12a2

Although our initial results indicate that Cas12a2 causes an abortive-infection phenotype, one scenario was that triggered Cas12a2 was selectively clearing all plasmids, enabling the cells to succumb to any introduced antibiotic selection. To assess this possibility, we evaluated the growth of *E. coli* in liquid culture under different antibiotic selection conditions (including a no-antibiotic condition) after induction of *Su*Cas12a2 and *Lb*Cas12a using a targeting crRNA (Fig. 4a and Supplementary Fig. 7). *Su*Cas12a2 but not *Lb*Cas12a suppressed culture growth in the absence of plasmid selection, further supporting induction of an abortive-infection phenotype by Cas12a2.

One open question is whether the abortive-infection phenotype was caused by cell dormancy or cell death. It was recently shown that, after recognizing an RNA target, Cas13a mediates widespread RNA degradation that drives cellular dormancy and suppresses phage infection[3]. We therefore introduced the representative Cas13a from *Leptotrichia shahii* (*Ls*Cas13a) into our liquid culture assay. Similar to *Su*Cas12a2, *Ls*Cas13a suppressed growth even in the absence of plasmid selection (Fig. 4a and Supplementary Fig. 7).

Our comparison to *Ls*Cas13a suggested that the growth suppression by *Su*Cas12a2 could occur through non-specific RNA cleavage, causing cell dormancy, whereas our in vitro data indicated that non-specific dsDNA cleavage could also suppress growth by causing cell death. To evaluate whether the cells containing *Su*Cas12a2 were undergoing cell death, we performed a cell-viability assay using propidium iodide for both Cas12a2 and Cas13a. We observed only a small percentage (about 10%) of cell death for both Cas12a2 and Cas13a after 4 h (Fig. 4b and Supplementary Fig. 8). Thus, although the indiscriminate nuclease activities of Cas12a2 cause some cell death, the primary result of Cas12a2 activity is better described as a cell-dormancy phenotype.

Although Cas12a2 appears to cause dormancy, it was unclear which of the several indiscriminate nuclease activities are involved. To determine whether *Su*Cas12a2 causes dormancy through RNA cleavage, total cellular RNAs were examined under targeting and non-targeting conditions. Whereas Cas13a significantly depleted both rRNAs and the small RNA pool (which includes tRNAs), Cas12a2 significantly depleted only the small RNA pool (Fig. 4c and Extended Data Fig. 7).

Given the observed differences in RNA degradation under targeting conditions, we examined whether the indiscriminate dsDNase activity of *Su*Cas12a2 was detectable in the context of the abortive-infection phenotype. We reasoned that widespread dsDNA damage caused by *Su*Cas12a2 would trigger an SOS response, impairing growth[40,41]. Consistent with this assertion, plasmid targeting using *Su*Cas12a2 significantly induced GFP expression from an SOS-responsive reporter

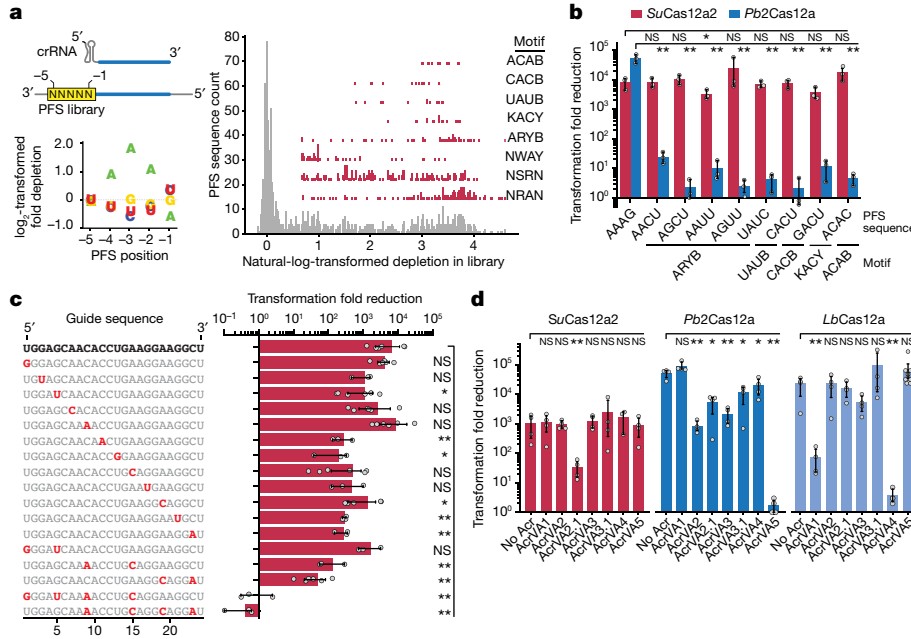

**Fig. 3 | *Su*Cas12a2 exhibits promiscuous targeting and resists anti-Cas12a proteins. a**, Experimentally determined PFSs and motifs recognized by *Su*Cas12a2 in *E. coli*. Motifs capturing positions −4 to −1 of the PFS are shown, and are written 3′ to 5′. B represents C, G or U; K represents G or U; R represents G or A; W represents A or U; and Y represents C or U. Results are representative of two independent screens (Extended Data Fig. 5). **b**, Validation of selected PFSs identified in the screen and permissive of targeting by *Su*Cas12a2 but not *Pb*2Cas12a. **c**, The effect of guide mismatches on plasmid targeting by *Su*Cas12a2 in *E. coli*. **d**, The extent of inhibition by known AcrVA proteins against *Su*Cas12a2. Acr proteins were confirmed to exhibit inhibitory activity against different Cas12a homologues in *E. coli* or in cell-free transcription–translation reactions (Extended Data Fig. 5). Data are mean ± s.d. of at least three independent experiments started from separate colonies. Statistical analysis was performed using one-tailed Welch's *t*-tests.

construct[42] compared with a non-target control, whereas *Lb*Cas12a and *Ls*Cas13a negligibly induced GFP expression (Fig. 4d and Supplementary Fig. 9). Furthermore, *Su*Cas12a2-targeting cultures diverged into two subpopulations in the absence of antibiotic selection: one represented by compact cells with reduced DNA content and another represented by filamentous cells with high DNA content (Fig. 4e and Extended Data Figs. 8 and 9). Cultures expressing *Lb*Cas12a and *Ls*Cas13a did not exhibit noticeable differences in cell size and DNA content for the target and non-target plasmids. Previous studies with other CRISPR–Cas systems that specifically targeted the bacterial chromosome observed similar morphological changes[43–45], suggesting that these distinct morphologies are due to dsDNA damage. These results demonstrate that RNA targeting by *Su*Cas12a2 causes dsDNA damage of the bacterial chromosome that in turn induces the SOS response and abortive infection in bacteria, reflecting a distinct mechanism of immunity that relies on indiscriminate dsDNase activity. Consistent with this observation, recent cryo-electron microscopy structures revealed that Cas12a2 binds to and cuts dsDNA through a mechanism that is completely distinct from all other CRISPR-associated nucleases, and structure-guided mutants with impaired in vitro collateral dsDNase but not ssRNase and ssDNase activities abolished in vivo defence activity against plasmids[19].

## RNA detection with Cas12a2

CRISPR single-effector nucleases have been repurposed for many applications from gene editing to molecular diagnostics. To determine whether Cas12a2 could be repurposed as a biotechnological tool, we co-opted *Su*Cas12a2 to detect RNA. We programmed apo *Su*Cas12a2 with a crRNA guide complementary to an RNA target and incubated the complex with a ssDNA or ssRNA beacon that fluoresces after cleavage due to separation of a fluorophore and a quencher (Fig. 4f). Using this approach, we were able to detect RNA using both ssDNA and ssRNA

probes at 37 °C and room temperature, with a limit of detection within the range observed for other single-subunit Cas nucleases[46] (Fig. 4f and Extended Data Fig. 10a–c). Furthermore, we devised a modified detection assay that uses plasmid DNA and DNA nick translation[47], introducing a distinct positive readout for CRISPR-based diagnostics (Extended Data Fig. 10d–f). These data indicate that Cas12a2 can be readily repurposed as a tool for applications in science, biotechnology, agriculture and medicine. We anticipate that the unique activities of this enzyme can be further leveraged to expand the CRISPR-based toolkit.

## Discussion

Collectively, our data support a model in which Cas12a2 nucleases exhibit RNA-triggered degradation of cytoplasmic dsDNA and small RNAs, impeding host cell growth and eliciting an abortive-infection phenotype (Fig. 4h). This mechanism contrasts with targeted invader clearance or abortive-infection activities exhibited by other CRISPR–Cas systems. Specifically, the mechanism exhibited by Cas12a2 is reminiscent of the recently described CBASS defence system, which relies on the indiscriminate double-stranded DNase NucC that degrades host cell DNA and kills the cell[12]. Notably, some type III systems encode NucC enzymes, suggesting that other CRISPR–Cas systems have convergently evolved to use a similar DNA-degrading abortive-infection mechanism[13,48].

In addition to damaging the genome and inducing the SOS response, *Su*Cas12a2 exhibits promiscuous RNA recognition through a flexible PFS and mismatch tolerance. This flexibility in target recognition mirrors the flexibility of tag–anti-tag complementarity observed with type III and VI systems[8,33] and could be particularly advantageous against rapidly evolving phages, although Cas12a2 orthologues must be characterized to determine whether such promiscuity is a common feature of these nucleases. Although flexible, the PFS would still prevent self-recognition of spurious

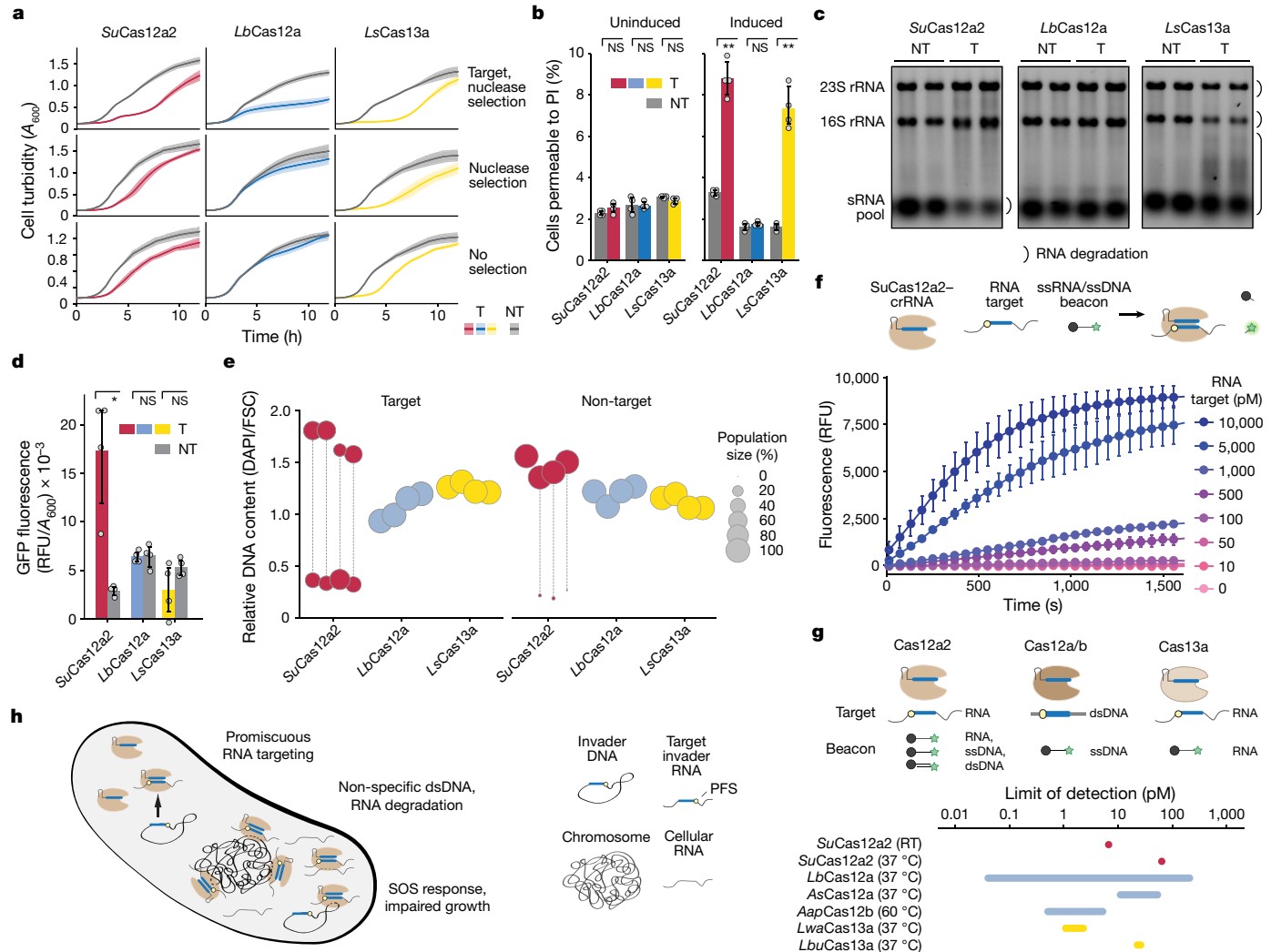

**Fig. 4 | *Su*Cas12a2 causes abortive infection principally through collateral DNA degradation and can be harnessed for RNA detection. a**, *E. coli* growth arrest in the presence of *Su*Cas12a2, *Lb*Cas2a2 or *Ls*Cas13a plasmid under different targeting conditions and antibiotic regimes. $A_{600}$, absorbance at 600 nm. **b**, The percentage of *E. coli* cells stained with propidium iodide (PI) indicative of viability loss before targeting (uninduced) and after 4 h of targeting (induced) without antibiotic selection. The gating strategy is shown in Supplementary Fig. 7. **c**, The varying extent of RNA degradation in *E. coli* by *Su*Cas12a2, *Lb*Cas12a or *Ls*Cas13a2 2 h after induction without antibiotic selection. The results represent duplicate independent experiments. See Extended Data Fig. 7 for independent quadruplicates. The small RNA pool includes tRNAs and other small RNAs. **d**, SOS-responsive expression of GFP in *E. coli* after 4 h of plasmid targeting by *Su*Cas12a2, *Lb*Cas12a or *Ls*Cas13a without antibiotic selection. Time-course data are shown in Supplementary Fig. 9. RFU, relative fluorescence units. **e**, Relative DNA content in *E. coli* after

4 h of targeting by *Su*Cas12a2, *Lb*Cas12a or *Ls*Cas13a without antibiotic selection. 4′,6-diamidino-2-phenylindole (DAPI) fluorescence and cell size were measured using flow cytometry. Each circle or pair of vertically aligned circles represents major subpopulations from the same biological replicate. The corresponding contour plots are shown in Extended Data Fig. 8. FSC, forward scatter. **f**, RNA-detection assay. The limit of RNA detection with Cas12a2 incubated with a ssRNA beacon at room temperature (RT). Data are mean ± s.e.m. of three independent experiments. **g**, Nucleic-acid targets and reporters as well as the unamplified limit of detection (LOD) for Cas12a2 and other Cas detectors[46]. *Aap*, *Alicyclobacillus acidiphilus*; *Lwa*, *Leptotrichia wadei*; *Lbu*, *Leptotrichia buccalis*. **h**, The proposed model for promiscuous RNA targeting and collateral degradation by *Su*Cas12a2 and its effect on the cell. For **b** and **d**, data are mean ± s.d. of four independent experiments started from separate colonies. Statistical analysis was performed using one-tailed Welch's *t*-tests. NT, non-target plasmid; T, target plasmid.

antisense transcription of the CRISPR array, as the corresponding PFS-containing portion of the repeat strongly diverges from the recognized PFS—a standard feature of self/nonself-recognition for PAMs in DNA-targeting CRISPR–Cas systems[49]. The flexibility in PFS and target recognition could further serve as a back-up mechanism if precise recognition and clearance of DNA targets by Cas12a fails in organisms that encode both Cas12a and Cas12a2 adjacent to a single CRISPR array (Fig. 1a,d and Extended Data Figs. 1 and 6b). This dual-nuclease strategy would be akin to bacteria encoding multiple CRISPR–Cas systems targeting the same invader[50]. However, further research is needed to understand how these two nucleases work together to counter infections.

The combination of nuclease-mediated crRNA biogenesis, RNA targeting and collateral cleavage of ssRNA, ssDNA and, in particular, dsDNA sets Cas12a2 apart from other known Cas nucleases. The apparent need to recognize the A-rich flanking sequence by *Su*Cas12a2 to activate the indiscriminate RuvC nuclease activity strongly indicates that Cas12a2 must bind to a correct PFS adjacent to the RNA target to activate cleavage rather than rely solely on complementarity between the repeat tag and target anti-tag pair to distinguish self from non-self sequences, typical of several other RNA-targeting Cas nucleases and complexes[8,33,51,52]. Investigating the underlying molecular basis of target recognition and activation of collateral cleavage by Cas12a2 could reveal new mechanisms used by CRISPR nucleases to discriminate

between self and non-self targets. Recent cryo-electron microscopy structures of Cas12a2 at stages of RNA targeting and collateral dsDNA capture are already fulfilling this need[19].

Cas12a2 holds substantial potential for CRISPR technologies. As a proof-of-principle demonstration, we showed that *Su*Cas12a2 can be repurposed for RNA detection with a limit of detection comparable to existing single-effector based tools[46]. Beyond the ability to detect RNA, we envision a variety of *Su*Cas12a2 applications that expand and enhance the CRISPR-based tool kit. RNA-triggered dsDNA cleavage could enable programmable killing of prokaryotic and eukaryotic cells with various applications, including programmable shaping of microbial communities, cancer therapeutics and counterselection to enhance genome editing. Moreover, the ability of Cas12a2 and Cas12a to use the same crRNA sequence yet recognize distinct nucleic acid species (RNA versus ssDNA and dsDNA) and elicit distinct non-specific cleavage activities (ssRNA, ssDNA and dsDNA (Cas12a2) versus ssDNA (Cas12a)) could augment existing Cas12a applications by incorporating Cas12a2. By further exploring the properties of *Su*Cas12a2 and its orthologues, we expect the advent of new and improved CRISPR technologies that could broadly benefit society.

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

## Methods

### Identification of the putative Cas12a2 nucleases
Several Cas12a2 sequences were initially identified and tentatively classified as encoding Cas12a nucleases[16]. These Cas12a2 protein sequences were used as seeds for BLASTp searches of protein data in NCBI and for tBLASTn searches of metagenomic data in NCBI (https://www.ncbi.nlm.nih.gov) and JGI (https://img.jgi.doe.gov) to identify additional putative Cas12a2 nucleases.

### Phylogenetic analysis of Cas12a2 proteins within type V systems
The amino acid sequences of Cas12a2, Cas12a and Cas13b orthologues were aligned using MAFFT (v.7.490)[53]. The resulting alignment was trimmed using ClipKIT[54] and used to create a maximum-likelihood phylogeny using RAxML-NG[55] with the following parameters: --model JTT+G --bs-metric fbp, tbe --tree pars{60}, rand{60} --seed 12345 --bs-trees autoMRE. Cas13b sequences were used as an outgroup. The amino acid sequences used in the creation of the phylogeny are provided in Supplementary File 1.

### Domain annotation and structure prediction
Conserved motifs in *Su*Cas12a2 were identified using MOTIF Search (https://www.genome.jp/tools/motif/, accessed on 15 June 2021) and Phyre 2[56] (accessed on 8 March 2021). HHpred secondary structure predictions of Cas12a2 orthologous amino acid sequences were performed to identify the common secondary structure between Cas12a2 and Cas12a that predicted the crRNA processing site of Cas12a2[57].

### Strains and plasmids
All of the in vivo experiments, unless indicated otherwise, were performed in *E. coli* BL21(AI). For propagation, the cultures were grown in LB medium at 37 °C with constant shaking at 225–250 rpm. *E. coli* strain TOP10 was used for plasmid cloning (Supplementary Table 1 (tab 1)). All primers, gBlocks and oligos were obtained from Integrated DNA Technologies, unless specified otherwise. Gibson assembly of plasmid construction was performed using the NEBuilder HiFi DNA Assembly Master Mix (New England Biolabs, E2621). Mutagenesis of the plasmids, including small insertions and nucleotide substitutions, was performed using the Q5 Site-Directed Mutagenesis Kit (New England Biolabs, E0554S). All of the nucleases together with crRNA, unless specified otherwise, were expressed from plasmids containing p15A origin-of-replication and a chloramphenicol-resistance marker. The expression of the nucleases and crRNA was controlled by a T7 promoter, unless otherwise specified. All of the target and non-target plasmids were created by introducing protospacer sequences and corresponding flanking sequences into pBR322 or sc101 origin-of-replication plasmids bearing a kanamycin-resistance cassette, unless otherwise specified. Sequences encoding Cas12a2 orthologues (Supplementary File 1) were codon-optimized and synthesized by Genscript. Sequences encoding *Pb*2Cas12a from *P. bryantii B14* (NCBI: WP_039871282) *Lb*Cas12a from *L. bacterium* ND2006 (NCBI: WP_035635841.1), *Fn*Cas12a from *Francisella tularensis* (NCBI: WP_104928540.1), *As*Cas12a from *Acidaminococcus* sp. BV3L6 (NCBI: WP_021736722.1) and *Mb*3Cas13a from *Moraxella bovoculi* (NCBI: WP_080946945.1)[16] were codon optimized for expression in *E. coli* and ordered as gBlocks from Integrated DNA Technologies. Sequences encoding anti-CRISPR proteins[36,38] (Supplementary Tables 1 and 3) were codon-optimized for expression in *E. coli* and ordered as gBlocks from Integrated DNA Technologies. The *acr* genes were then PCR-amplified and introduced into the pBAD24 plasmid backbone carrying an ampicillin-resistance cassette[58]. The *Ls*Cas13a-encoding plasmid pCBS2091 was ordered from Addgene (79150)[8]. For detecting RecA-dependent SOS response in *E. coli* BL21(AI), reporter plasmids pCBS2000, pCBS3611 and pCBS3616 were created by introducing the *recA* promoter, included 100 bp upstream of the predicted LexA-binding site, upstream of the GFP-encoding gene

into plasmid pCBS198. Plasmids pCBS3611 and pCBS3616 received an ampicillin-resistance cassette from plasmid pCB672[24]. The *recA* promoter sequence was identified in the genome of *E. coli* BL21(AI) between positions 2,635,525 and 2,635,347 (NCBI: CP047231.1). Control plasmids pCBS3616 and pCBS2002 without the GFP-reporter genes were generated by PCR amplification of pCBS2000 and pCBS3616 followed by KLD assembly (New England Biolabs, M0554). A full list of plasmids used in the study, including links to plasmid maps, is provided in Supplementary Tables 1 and 2. A list of relevant oligonucleotide, dsDNA and RNA sequences is provided in Supplementary Tables 1 and 3.

### In vitro characterization of *Su*Cas12a2
**Expression and purification of *Su*Cas12a2.** N-terminal 6×His-tagged *Su*Cas12a2 WT and mutant constructs were expressed in *E. coli* Nico21(DE3) cells from a pACYC plasmid either lacking (apo) (plasmid 1416) or containing a three-spacer CRISPR array (crRNA-guided) (plasmid 1408) using either autoinduction or isopropyl β-D-1-thiogalactopyranoside (IPTG) induction. Autoinduction growth was performed according to previously reported guidelines[59]. In brief, a solution containing recommended concentrations of ZY media, $MgSO_4$, metals mix, 5052 (0.5% glycerol, 0.05% glucose, 0.2% α-lactose) and NPS autoinduction buffers along with antibiotics needed for selection was inoculated with bacteria from a glycerol stock or a fresh transformation. The cells were grown for 5 h at 37 °C with shaking at around 250 rpm and then moved to 24 °C where they were incubated for 24 h before collection by centrifugation at 8,000 rpm for 25 min. Cell pellets were then stored at −80 °C until purification. For the IPTG induction, 1 l of TB medium was inoculated with 20 ml of overnight growth and was grown at 37 °C until an optical density at 600 nm ($OD_{600}$) of 0.6. The cells were then cold-shocked on ice for 15 min and induced with 0.1 mM IPTG, followed by a 16–18 h incubation at 18 °C. Cells were collected by centrifugation. Cells were lysed by sonication in lysis buffer (25 mM Tris pH 7.2, 500 mM NaCl, 10 mM imidazole, 2 mM $MgCl_2$, 10% glycerol) in the presence of leupeptin, aprotinin, pepstatin, AEBSF and lysozyme. The lysate was clarified by centrifugation at 36,400$g$ for 35 min. Clarified lysate was added to 5 ml of Ni-NTA resin and batch bound at 4 °C for 30 min, and then washed with 100 ml of lysis buffer. The protein was eluted with 50 ml of Ni-elution buffer (25 mM Tris pH 7.2, 500 mM NaCl, 250 mM imidazole, 2 mM $MgCl_2$, 10% glycerol). Fractions containing *Su*Cas12a2 were desalted using the Hiprep 26/10 desalting column into low-salt buffer (25 mM Tris-pCas12a22, 50 mM NaCl, 2 mM $MgCl_2$, 10% glycerol). *Su*Cas12a2 + crRNA was then applied to a Hitrap Q HP column anion-exchange column, whereas the apo *Su*Cas12a2 was applied to a Hitrap SP HP cation-exchange column. The column was washed with 10% high-salt buffer (25 mM Tris pH 7.2, 1 M NaCl, 2 mM $MgCl_2$, 10% glycerol) followed by a gradient elution to 100% high-salt buffer 10 CV (50 ml). The fractions containing *Su*Cas12a2 were concentrated using a 100 kDa MWKO concentrator to about 1 ml and then purified by size-exclusion column chromatography over a Hiload 26/600 Superdex 200 pg column equilibrated in SEC buffer (100 mM HEPES pH 7.2, 150 mM KCl, 2 mM $MgCl_2$, 10% glycerol). The fractions containing *Su*Cas12a2 were concentrated and stored at −80 °C.

**Pre-crRNA processing.** For processing of a 3× pre-crRNA, *Su*Cas12a2 pre-CRISPR×3 RNA was in vitro transcribed using the HiScribe T7 High Yield RNA Synthesis Kit (New England Biolabs). The template DNA was derived from Jackson Laboratory plasmid 1409 linearizing with the KpnI restriction enzyme. A contaminating band that runs approximately at 130 nucleotides was observed to be an artifact of the reaction. Numerous strategies were attempted to prevent the transcription of this contaminating band, to no success. In vitro transcribed RNA was cleaned using RNeasy spin columns (Qiagen). Then, 1.5 µM of apo *Su*Cas12a2 was incubated with 1 mg of *Su*Cas12a2 pre-CRISPR×3 RNA in 1× 3.1 buffer from New England Biolabs (100 mM NaCl, 50 mM Tris-HCl, 10 mM $MgCl_2$, 100 mg ml$^{-1}$ BSA pH 7.9) and incubated at 25 °C for various

times. The samples were run on a gel (12% polyacrylamide, 8 M, TBE) alongside a ssRNA low range ladder (New England Biolabs) and stained with SYBR gold (Thermo Fisher Scientific).

For processing of a 1× crRNA with WT and crRNA-processing mutants, a synthetic crRNA with a 13 base 5′ unprocessed overhang (smcrRNA; Supplementary Tables 1 and 3) was refolded using a previously outlined protocol[60]. In a 10 µl reaction, 150 nM of crRNA substrate was combined with 1.5 µM WT, K784A or K785A apo SuCas12a2 protein in NEB 3.1. The reactions were incubated at 37 °C for 1 h. The reactions were quenched with phenol and phenol−chloroform extraction was performed. The results were analysed using 12% urea−PAGE stained with SYBR Gold.

**Nucleic acid cleavage assays.** For analysis of targeted cleavage, 10 µl reactions of 250 nM SuCas12a2−crRNA with 100 nM of complementary FAM-labelled synthetic oligonucleotide (that is, ssDNA, dsDNA or RNA) in 1× NEB 3.1 buffer were incubated at 37 °C for 1 h. Reactions were quenched with phenol and then phenol−chloroform extraction was performed. Results were analysed using a previously outlined FDF-PAGE method[61] and visualized for fluorescein fluorescence.

For analysis of collateral cleavage, 10 µl reactions of 250 nM SuCas12a2−crRNA, and 250 nM of target (RNA complementary to the crRNA-guide) or non-target (RNA non-complementary to the crRNA-guide) substrate and 100 nM of 5′-FAM labelled collateral substrate (ssDNA, dsDNA, RNA) in 1× NEB 3.1 were incubated at 37 °C for 1 h. Reactions were quenched with phenol and then phenol−chloroform extraction was performed. The results were analysed using 12% urea−PAGE and visualized for fluorescein fluorescence.

For analysis of the flanking-sequence requirements for activation, 10 µl reactions of 250 nM Cas12a2−crRNA, with 300 nM of different target ssRNAs (self (flanked by sequence complementary to the direct repeat of the crRNA), no flanks and flanks containing a 5′-GAAA-3′ PFS on the 3′ side of the protospacer) and 100 nM of collateral 5′-FAM dsDNA in 1× NEB 3.1 buffer were incubated at 37 °C for 1 h. The reactions were quenched with phenol and phenol−chloroform extraction was performed. The results were analysed using 12% urea−PAGE and visualized for fluorescein fluorescence.

For kinetic analysis of collateral cleavage, a single 100 µl reaction containing 100 nM Cas12a2−crRNA, 100 nM of target ssRNA (crRNA complementary) and 100 nM of different 5′-FAM labelled collateral substrates (ssDNA, dsDNA, RNA) in 1× NEB 3.1 buffer was made. Time points were taken at 1, 2, 5, 10, 15, 30, 60, 120 and 180 min by combining 10 µl from the 100 µl reaction with phenol, and then performing phenol−chloroform extraction. The results were analysed using 12% urea−PAGE and visualized for fluorescein fluorescence.

For the plasmid-cleavage assay, a 100 µl reaction containing 14 nM Cas12a2−crRNA, 25 nM target RNA, 7 nM of pUC19 plasmid in 1× NEB 3.1 buffer was incubated at 37 °C. At the indicated time points, 10 µl of the reaction was removed and quenched with phenol and phenol−chloroform extraction was performed. The reactions were visualized on 1% agarose with ethidium bromide.

**Collateral cleavage comparison between Cas12a2 and Cas12a, Cas13a and Cas12g.** EnGen LbaCas12a (LbCas12a) was purchased from New England Biolabs (M0653S). Reactions (10 µl) containing 250 nM of LbCas12a and 500 nM of its cognate crRNA in 1× NEB 2.1 buffer were incubated at 37 °C with 200 nM of different target substrates (ssDNA, dsDNA, RNA) and 100 nM of different FAM-labelled collateral substrates (ssDNA, dsDNA, RNA). After 1 h, the reactions were quenched by phenol and phenol−chloroform extraction was performed. The results were analysed using 12% urea−PAGE and visualized for fluorescein fluorescence.

LwCas13a was purchased from MCLAB Molecular Cloning Laboratories (Cas13a-100). Reactions (10 µl) containing 250 nM of LwCas13a and 500 nM of its cognate crRNA in the provided 1× Cas9 buffer

(20 mM HEPES (pH 6.5), 5 mM MgCl2, 100 mM NaCl, 100 µM EDTA) were incubated at 37 °C with 200 nM of different target substrates (ssDNA, dsDNA, RNA) and 100 nM of different FAM-labelled collateral substrates (ssDNA, dsDNA, RNA). After 1 h, the reactions were quenched by phenol and phenol−chloroform extraction was performed. The results were analysed using 12% urea−PAGE and visualized for fluorescein fluorescence.

AbCas12g was expressed in E. coli NiCo 21 DE3 using pET28a-mH6-Cas12g1 (Addgene plasmid, 120879) and initially purified as described previously[25]. The protein was then transferred to low-salt buffer (25 mM HEPES pH 7.8, 50 mM NH4Cl, 2 mM MgCl2, 7 mM BME, 5% glycerol) by buffer-exchange and loaded over heparin followed by elution with a linear NaCl gradient and gel-filtration as described previously[62]. Purified protein was flash-frozen and stored at −80 °C. The Cas12g1 non-coding plasmid pACYC-Cas12g1 (Addgene plasmid, 120880) was used as a template for PCR amplification of the AbCas12g tracrRNA sequence with Cas12gtracrRNA F and R primers (Supplementary Tables 1 and 3) in 2× Taq Master Mix (New England Biolabs). The non-coding plasmid was removed with DpnI by incubation at 37 °C for 1 h in CutSmart buffer (New England Biolabs). DNA components were cleaned after PCR and DpnI digestion using the E.Z.N.A. Cycle Pure Kit (OMEGA BioTek). The Cas12g tracrRNA was transcribed using the HighScribe T7 Quick High Yield RNA synthesis kit and cleaned using the Monarch RNA cleanup kit (New England Biolabs). Reactions (10 µl) containing 250 nM of Cas12g, 500 nM of the Cas12g crRNA and 1 µM of Cas12g tracrRNA in 1× NEB 3.1 buffer were incubated at 37 °C or 50 °C with 200 nM of different target substrates (ssDNA, dsDNA, RNA) and 100 nM of different FAM-labelled collateral substrates (ssDNA, dsDNA, RNA). After 1 h, the reactions were quenched by phenol and phenol−chloroform extraction was performed. The results were analysed using 12% urea−PAGE and visualized for fluorescein fluorescence.

To analyse SuCas12a2 collateral activity, 10 µl reactions containing 250 nM of Cas12a2−crRNA, 200 nM of different target substrates (ssDNA, dsDNA, ssRNA) and 100 nM of different FAM-labelled collateral substrates (ssDNA, dsDNA, RNA) in 1× NEB 3.1 buffer were incubated at 37 °C for 1 h. The reactions were quenched with phenol and phenol−chloroform extraction was performed. The results were analysed using 12% urea−PAGE and visualized for fluorescein.

**RNA detection by Cas12a2 with ssRNA and ssDNA reporter probes.** Cas12a2 (100 nM) was complexed with crRNA (120 nM) in NEB 3.1 buffer (50 mM Tris-HCl pH 7.9, 100 mM NaCl, 10 mM MgCl2, 100 µg ml⁻¹ BSA) before combining with RNase or DNase Alert (200 nM, IDT) and Target RNA to the indicated concentrations in a 384-well plate (Greiner Bio-One, 784077). A background control was prepared with nuclease-free water instead of target RNA. The reactions were monitored for reporter fluorescence (RNase Alert: excitation 485-20/ emission 528-20, DNase Alert: excitation 500-20/emission 560-20) over time at either ambient conditions (room temperature) or 37 °C using the Synergy H4 Hybrid multi-mode microplate reader (BioTek Instruments). The slope of the linear region (between 5 and 30 min) was determined at each concentration of target RNA using GraphPad PRISM. Standard error of the linear fit was used as a proxy for standard deviation, and the limit of detection was calculated as 3 × standard error of the water background as described previously[46]. The limit of detection was estimated by determining where the plot of $V_0$ (relative fluorescence units (RFU)/s) versus concentration of target RNA crosses the detection threshold.

**Plasmid cleavage.** Plasmid cleavage reactions were prepared by combining 14 nM SuCas12a2 (or mutant) with 14 nM crRNA and 25 mM Target RNA in NEB 3.1 buffer (50 mM Tris-HCl pH 7.9, 100 mM NaCl, 10 mM MgCl2, 100 µg ml⁻¹ BSA). Protein was preheated at 37 °C for 15 min before the addition of 7 nM supercoiled pUC19 plasmid. Samples were removed at the time points 1, 2, 5, 10, 20, 30, 45 and 60 min and

quenched in pH 8.0 phenol–chloroform. Quenched reactions were mixed by flicking followed by centrifugation. The samples were loaded onto 1% agarose gels and visualized using ethidium bromide. Gels were imaged using the ChemiDoc MP Image System (Bio-Rad).

**Nick translation.** The plasmid pSPC421 was collected from TOP10 *E. coli* cells using the ZymoPURE II Plasmid Midiprep Kit by Zymo Research (D4201), and cleaned using the DNA Clean & Concentrator-5 kit from Zymo Research (D4013). ca33Cas12a2 was expressed from the plasmid pCBS5042 and purified as described above at the Rudolf Virchow Center for Integrative and Translational Bioimaging. ca33Cas12a2 nuclease (100 nM) was incubated with the crRNA (1 µM) in the NEB3.1 buffer for 30 min at room temperature. CAO1 target RNA (1 nM) and pSPC425 (3 µg) were added to the reaction medium for 15 min. For evaluating plasmid nicking, the samples were heated at 80 °C for between 1 and 30 min. The reactions were run on a 0.8% agarose gel. DNA polymerase I (NEB, M0209L) was added to the reaction (0.2 U µl$^{-1}$) with the Atto421-NT labelling mix (1×) and NT labelling buffer (1×) from the Atto425 NT Labelling Kit (Jena Bioscience, PP-305S-425). The samples were incubated at 15 °C for 90 min in a Bio-Rad thermocycler. The resulting labelled DNA fragments were purified using the Microspin S-400 HR columns (Cytiva, 27514001). Fluorescence measurements ($\lambda_{exc}$ = 436 nm; $\lambda_{em}$ = 484 nm) were performed on a fluorescence microtitre plate reader (BioTek NeoG2) at 25 °C.

### Cas12a2 characterization in *E. coli*

**crRNA sequencing and analysis.** The *Su*Cas12a2 expression plasmid pCBS3568 containing the nuclease- and the crRNA-encoding sequences and the no-crRNA control pCBS3569 were transformed into *E. coli* BL21(AI) and the transformants were plated on selection pates. The resulting colonies were picked and used to inoculate 2 ml overnight liquid cultures. The next day, the overnight cultures were used to inoculate 25 ml of LB containing chloramphenicol to an OD$_{600}$ of approximately 0.05. Once the growing cultures reached an OD$_{600}$ of 0.25 after approximately 40 min, expression of the nuclease and the crRNA were induced with 1 mM IPTG and 0.2% L-arabinose. The induced cultures were collected in the stationary phase by centrifugation at 14,000 rpm at 4 °C for 2 min. The cell pellets were then immediately frozen in liquid N$_2$ and stored at −80 °C until further processing.

Total RNA was purified from cell pellets using the Direct-zol RNA Miniprep Plus (Zymo Research, R2072) according to the manufacturer's instructions. DNA was removed using TurboDNase (Life Technologies, AM2238). Between the individual processing steps, RNA was purified using the RNA Clean & Concentrator kit (Zymo Research, R1017). Ribosomal RNA was removed from the samples using the RiboMinus Transcriptome Isolation Kit, bacteria (Thermo Fisher Scientific, K155004). 3′-phosphoryl groups were removed from RNA using T4 polynucleotide kinase (New England Biolabs, M0201S). cDNA synthesis and library preparation was performed using the NEBNext Multiplex Small RNA Library Prep Set for Illumina (New England Biolabs, E7330S). Size selection for fragments between 200 bp and 700 bp was performed using the Select-a-Size DNA Clean & Concentrator kit (Zymo Research, D4080). Finally, DNA was purified using AMPure XP beads (Beckman Coulter, A63882) and quantified using the Qubit dsDNA HS assay kit (Thermo Fisher Scientific, Q32851) on DeNovix DS-11 FX (DeNovix).

Library sequencing was performed at the Helmholtz Center for Infectious Research (HZI) GMAK facility in Braunschweig, Germany, using the MiSeq 300 sequencing method (Illumina). The resulting paired-end reads were quality controlled, trimmed and merged using BBTools[63] (https://sourceforge.net/projects/bbmap/). Next, the reads were mapped to the crRNA expression site on the plus strand of pCBS273 using Bowtie2 (http://bowtie-bio.sourceforge.net/bowtie2/). The associated raw and processed sequencing data as well the data-processing steps can be found at the NCBI Gene Expression Omnibus (GEO: GSE178531).

**Plasmid clearance assay in *E. coli*.** Standard plasmid clearance assays were performed in *E. coli* BL21(AI) containing nuclease- and crRNA-expressing plasmids. Bacterial cultures were grown overnight and used to inoculate fresh LB medium containing chloramphenicol to an OD$_{600}$ of 0.05–0.1. Subsequently, these cultures were grown until the OD$_{600}$ reached approximately 0.25, at which time 1 mM IPTG and 0.2% L-arabinose were added for induction. Once the cultures reached an OD$_{600}$ of 0.6–0.8, the cells were collected and made electrocompetent[64]. Electrocompetent cells were prepared from four biological replicates. Immediately after, 1 µl of 50 ng µl$^{-1}$ of the target and non-target plasmid were electroporated into 50 µl of the electrocompetent *E. coli* cells. To achieve high transformation efficiencies, the used plasmids were purified through ethanol precipitation and quantified using the Qubit dsDNA HS Assay Kit (Thermo Fisher Scientific, Q32851). The electroporated cells were recovered for 1 h at 37 °C with shaking in 500 µl LB containing 1 mM IPTG and 0.2% L-arabinose without antibiotics. Next, the cultures were sequentially diluted to 10$^{-5}$ in tenfold increments. Then, 5–10 µl of each dilution was spotted onto LB plates containing antibiotics to select the nuclease-crRNA and the target/non-target plasmids. The plates also contained 0.3 mM IPTG and 0.2% L-arabinose. The plates were incubated overnight at 37 °C.

The next day, the colonies were manually counted and the resulting counts were adjusted for the dilution factor. Counts from the highest countable dilution were used to calculate transformation fold reduction as a ratio between the colonies in the non-target condition divided by the colonies in the target condition.

In a modification of the assay used to determine the cell suicide phenotype, the target and the non-target plasmids were transformed into *E. coli* BL21(AI) first. Next, these cells were made electrocompetent and the nuclease-crRNA plasmids were transformed in last.

When testing Acrs, the Acr plasmid (ampicillin) and the nuclease-crRNA plasmid (chloramphenicol) were co-transformed, followed by electroporation of the target or non-target plasmid (kanamycin).

**Growth experiments.** To investigate the growth of the cultures under nuclease-targeting conditions, the nuclease-crRNA and the target/non-target plasmids were transformed into *E. coli* BL21(AI). The resulting transformants were recovered in SOC medium and grown overnight with 0.2% glucose to inhibit nuclease and crRNA expression. In the morning, the cells were collected by centrifugation at 5,000*g* for 2 min. The pellets were resuspended in LB and used to inoculate 200 µl of LB medium on a 96-well plate to a final OD$_{600}$ of 0.01. Depending on the experiment, the reactions contained different combinations of antibiotics, IPTG and L-arabinose. The plates were incubated in the BioTek Synergy H1 plate reader at 37 °C with vigorous shaking. The OD$_{600}$ of the cultures was recorded every 3 min. Plasmid clearance assays were performed with the overnight cultures, as described above.

**PFS depletion assay in *E. coli*.** To determine PFS preferences of *Su*Cas12a2, a PFS depletion assay was performed. An oligo library (ODpr23) consisting of 1,024 nucleotide combinations in place of a 5-nucleotide PFS-encoding site was synthesized by Integrated DNA Technologies. Using the ODpr23 oligo pool library in a combination with primer ODpr24, targeting plasmid pCBS276 was PCR-amplified using Q5 polymerase (New England Biolabs, M0543). The PCR products were gel-purified using the Zymoclean Gel DNA Recovery Kit (Zymo Research, D4007) and ligated using the KLD reaction mix (New England Biolabs, M0554). The ligated plasmids were purified using ethanol precipitation and electroporated into *E. coli* TOP10. A total of ten electroporation reactions were performed. After recovery of the electroporated cells in SOC medium, the individual reactions were combined to inoculate 90 ml of LB medium containing kanamycin. A total of 10 µl from each electroporation reaction was plated on selective LB medium to estimate the total number of transformed bacteria.

With the colony counts, we estimated that the total number of transformed cells exceeded the number of unique PAM sequences in the library (1,024) by approximately 2,300-fold. Plasmid library DNA was purified from the combined overnight culture using the ZymoPURE II Plasmid Midiprep Kit (Zymo Research, D4201) and additionally cleaned by ethanol precipitation. Next, the plasmid library was verified by Sanger sequencing.

The PAM plasmid library was transformed into electrocompetent *E. coli* BL21(AI) containing either the *Su*Cas12a2 nuclease-expressing plasmid pCBS273 or an empty plasmid control pCBS3569. The electrocompetent cells were prepared as described above. Approximately 600 ng of the plasmid DNA was electroporated into 50 µl volume of the competent cells. The transformed bacteria were recovered in 500 µl of SOC medium for 1 h at 37 °C and were used to inoculate 50 ml LB with 1 mM IPTG and 0.2% L-arabinose in the presence of kanamycin and chloramphenicol. The cultures were grown for 13 h before the cells were collected by centrifugation at 4,000*g* for 15 min and the plasmid DNA extracted using ZymoPURE II Plasmid Midiprep Kit (Zymo Research, D4201). After recovery, bacteria were also plated on LB plates containing kanamycin and chloramphenicol without the inducers. These plates were used to estimate the total number of cells transformed with the plasmid library. The total number of transformed cells estimated based on the colony counts exceeded the number of unique PAM sequences in the library by approximately 1,700-fold for the cells containing the *Su*Cas12a2–crRNA plasmid (pCBS273) compared to 11,900-fold in the no-crRNA control (pCBS3569).

The region of the plasmid DNA containing the target site including the PFS-encoding sequence was PCR-amplified using the primers ODpr55 and ODpr56. The PCR reactions were purified using AMPure XP beads (Beckman Coulter, A63882). The purified PCR products were indexed using the primers ODpr58, ODpr60, ODpr59 and ODpr61. The indexed PCR products were purified using the AMPure XP beads, quantified using the Qubit assay (Thermo Fisher Scientific, Q32851) and sent for sequencing at the HZI GMAK facility using the MiSeq PE300 Illumina sequencing method.

Analysis of the PFS-encoding sequence depletion data as well as the creation of the PFS wheels were performed as described previously[65]. PFS consensus motifs were defined manually. The raw and the processed sequencing data as well as the data-processing steps can be found at the NCBI GEO (GSE178530). Individual PFS sequences were validated using plasmid-clearance assays as described above.

**Cell-free transcription–translation reactions.** For in vitro assays to test Acr sensitivity of Cas12a nucleases, plasmids encoding Cas12a nuclease were pre-expressed together with a plasmid encoding either a target or non-target crRNA in 9 µl of MyTXTL master mix (Arbor Biosciences) at a final concentration of 4 nM for each plasmid in a total volume of 12 µl. Acrs were pre-expressed separately, at a concentration of 4 nM in a total volume of 12 µl. As the Acrs are encoded on linear DNA fragments, GamS at a final concentration of 2 µM was added to prevent DNA degradation. All pre-expressions were performed at 29 °C for 16 h. The subsequent cleavage assay was performed by adding 1 µl of each pre-expression reaction to 9 µl of fresh myTXTL mix. pCBS420 plasmid constitutively expressing deGFP protein was used as a reporter at a final concentration of 1 nM. For quantification, four 3 µl replicates per reaction were transferred onto a 96-well V-bottom plate (Corning Costar 3357). The reactions were prepared using the Echo 525 Liquid Handler (Beckman Coulter). Fluorescence was measured on the BioTek Synergy H1 plate reader (excitation, 485/20; emission, 528/20). Time-course measurements were run for 16 h at 29 °C, with 3 min intervals between the measurements.

All fold-repression values for plasmid reporter constructs represent the ratio of deGFP concentrations after 16 h of reaction for the non-target over the target crRNA. For the experiments measuring the inhibitory activity of Acrs, inhibition was calculated from end-point expression values after 16 h of expression according to the following formula[66]: percentage inhibition of nuclease activity = $100 \times (RFU_{t,Acr}/RFU_{nt,Acr} - RFU_{t,-}/RFU_{nt,-})/(1 - RFU_{t,-}/RFU_{nt,-})$, where the inhibition of nuclease activity (%) is defined by the ratio of fluorescence between GFP targeting (t) and non-targeting (nt) Cas nucleases in the presence (Acr) and absence (-) of Acrs.

**Quantification of SOS response.** To measure the RecA-dependent SOS response, the nuclease-crRNA, the target/non-target plasmids (pCBS276/pCBS3578, kanamycin) and the reporter PrecA-gfp/no-gfp (pCBS3611/pCBS3616, ampicillin) plasmids were transformed into *E. coli* BL21(AI) sequentially. The plasmids pCBS273 and pCBS3588 (chloramphenicol) were used to express *Su*Cas12a2 and *Lb*Cas12a nucleases, respectively. When measuring the RecA-dependent SOS response in the presence of *Ls*Cas13a, the nuclease expression plasmid pCBS361 (chloramphenicol) was used. The target/non-target plasmids pCBS2004/pCBS612 (ampicillin) and PrecA-gfp/no-gfp plasmids pCBS2000/pCBS2002 (kanamycin) were used. First, the cells were grown in LB medium with 0.2% glucose to inhibit the expression of the nucleases and the crRNA. The bacteria were collected from the overnight cultures (15 ml) by centrifugation at 5,000*g* for 2 min and resuspended in fresh LB. Next, 200 µl of fresh LB medium was inoculated onto 96-well plates with the resuspended bacteria from the overnight cultures. These cultures were grown in the presence of either chloramphenicol, kanamycin and ampicillin, chloramphenicol and ampicillin, or no antibiotics. For the induction of nuclease and crRNA expression 1 mM of IPTG and 0.2% L-arabinose were added.

The cultures were grown at 37 °C with vigorous shaking. $OD_{600}$ and fluorescence measurements (excitation: 485/20, emission: 528/20) were collected every 5 min on the BioTek Synergy H1 plate reader. Four biological replicates were measured per experimental condition.

To determine whether a change in fluorescence occurred as a result of nuclease targeting, first the background fluorescence collected for the cultures with the PrecA-no-gfp plasmid (pCBS3616/pCBS2002) was subtracted from the values obtained for the cultures with the GFP-expressing plasmids for each time point (pCBS3611/pCBS2000). Next, the fluorescence values were divided by the $OD_{600}$ values from the corresponding target and the non-target cultures. Statistical significance was determined using Welch's *t*-test with unequal variance.

In parallel we performed a plasmid-clearance assay with the washed overnight cultures (Supplementary Fig. 9b), as described above. For the lowest plated dilution, cultures at an $OD_{600}$ of around 0.1 were used.

**Flow cytometry.** For the flow cytometry measurements, *E. coli* BL21(AI) cells were sequentially electroporated with the nuclease-encoding and target/non-target plasmids. The *Su*Cas12a2- and *Lb*Cas12a-expressing plasmids pCBS273 and pCBS3588 were used, respectively. Target plasmid pCBS273 and non-target plasmid pCBS3578 were used. For the experiments involving *Ls*Cas13a, nuclease-expression plasmid pCBS361 was used in combination with the target plasmid pCBS2004 and non-target plasmid pCBS612. After plasmid transformation, the *E. coli* bacteria were recovered in SOC medium and grown overnight in LB with chloramphenicol, kanamycin and 0.2% glucose. Next, the cells were collected at 5,000*g* for 2 min and resuspended in fresh LB. The resuspended bacteria were used to inoculate 15 ml cultures to an $OD_{600}$ of about 0.01. These cultures were grown at 37 °C with 220 rpm shaking for 6 h without antibiotics with 1 mM IPTG and 0.2% L-arabinose. Every 2 h the $OD_{600}$ of the cultures was measured and 500 µl samples were collected and centrifuged for 3 min at 5,000*g*. The cell pellets were then resuspended in 1× PBS containing 2 µg ml⁻¹ DAPI (Thermo Fisher Scientific, 62248). The resuspended cells were stained for 10 min in the dark, after which 10 µl was transferred into 240 µl of 1× PBS on a 96-well plate. DAPI fluorescence was measured using the Cytoflow Novocyte Quanteon flow cytometer as emission in the Pacific Blue

spectrum (455 nm). Data regarding the forward scatter (FSC) and the side scatter were also collected.

The resulting data were analysed in Python. First, clusters of bacteria exhibiting distinct FSC and Pacific Blue signals were identified using density-based spatial clustering of applications with noise (DBSCAN; https://scikit-learn.org/stable/modules/generated/sklearn.cluster.DBSCAN.html). Next, the ratios of the Pacific Blue to the FSC signal for each data point and the percentage of the data points within each cluster were parsed from the clustering data. The resulting values were plotted in the form of balloon plots. A total of 60,000 events per sample were analysed.

**Dead/live staining.** Dead and viable bacteria were estimated using the LIVE/DEAD BacLight Bacterial Viability and Counting Kit (Molecular Probes, L34856). The measurements were performed using the Cytoflow Novocyte Quanteon flow cytometer. *E. coli* BL21(AI) bacteria were transformed with nuclease, crRNA, and either target or non-target expression plasmids. For expressing *Su*Cas12a2 and *Lb*Cas12a with a target guide, the plasmids pCBS273 and pCBS3588 were used, respectively. Target expression plasmid pCBS2004 and non-target expression plasmid pCBS612 were used. For expressing *Ls*Cas13a with target and non-target guides, plasmids pCBS273 and pCBS3578 were used, respectively. Cultures containing combinations of nuclease–guide and target plasmids were grown for approximately 16 h with 0.2% glucose inhibitor in four biological replicates. Next, 1 ml of each culture was collected by centrifugation at 5,000$g$ for 3 min. The resulting pellet was resuspended in 1 ml of fresh LB medium. A total of 60 μl of this suspension was used to inoculate 20 ml of LB. Three cultures were grown for 2 h at 37 °C with constant shaking at 220 rpm. The expression of the nucleases and the guides was induced with 0.2% arabinose and 0.01 mM IPTG. After 4 h, the OD$_{600}$ of the cultures was measured. A volume of the cultures corresponding to an OD$_{600}$ of 1.0 was collected and processed as described in the kit manual. In brief, samples of the bacterial culture were centrifuged at 10,000$g$ for 3 min to pellet the cells. The supernatant was removed and the pellet was resuspended in 1 ml of 0.85% NaCl. As a control for the dead cells, spectate pellet was first resuspended in 300 μl 0.85% NaCl and then 700 μl 70% isopropyl alcohol (dead-cell suspension). The samples were incubated at room temperature for 60 min, with mixing every 15 min. Next, the samples were centrifuged at 10,000$g$ for 3 min and washed in 1 ml 0.85% NaCl, followed by another centrifugation. Finally, the samples were resuspended in 0.5 ml of 0.85% NaCl. One millilitre of the master mix for staining the cells contained 977 μl of 0.85% NaCl, 1.5 μl of component A (3.34 mM SYTO 9 nucleic acid stain), 1.5 μl of component B (30 mM propidium iodide (PI)), 10 μl of component C (beads) and 10 μl of the sample. These reactions were incubated for 15 min at room temperature protected from light. Fluorescence was collected in the green (fluorescein for SYTO 9) and red (Texas Red for PI) channels. The dead cells in each sample were gated on the basis of the dead-cell suspension control treated with isopropyl alcohol. The percentage of dead cells stained with PI was calculated from the total number of events without the beads. A total of 50,000 events were counted per sample.

**In vivo RNA degradation.** Samples corresponding to 1 ml of culture at an OD$_{600}$ of 0.4 grown for dead/live staining, as described above, were collected and centrifuged at 10,000$g$ for 3 min. The resulting pellets were frozen in liquid nitrogen and stored −80 °C until further processing. Total RNA was extracted using 1.5 ml of Trizol and 1.5 ml of ethanol with the Direct-zol RNA Miniprep kit (R2051, Zymo), according to the manufacturer's instructions. The RNA was further purified using the RNA Clean & Concentrator-5 kit (R1013, Zymo). A total of 0.5 μg of RNA from each sample in 5 μl was combined with 2.5 μl of RNA loading dye, heated to 70 °C for 10 min and subsequently chilled on ice for 2 min. The RNA High-Range ladder that was used was also heat-treated. The denatured samples (5 μl) and the leader (3 μl) were loaded onto

a 1% TBE gel. The gel was run for 40 min at 120 V. Next, the gel was stained for 30 min in ethidium bromide, washed for 10 ml and imaged. Gel images were analysed using GelAnalyzer v.19.1 (www.gelanalyzer.com).

**Microscopy.** For confocal microscopy, the cells were grown as described above for flow cytometry. At 2 h intervals, 500 μl of each culture was collected and centrifuged at 5,000$g$ for 3 min. Next, the bacteria were diluted to approximately the same cell density and stained with 2 μg ml$^{-1}$ of FM4-64 dye (Thermo Fisher Scientific, T13320) and 1 μg ml$^{-1}$ of DAPI (Thermo Fisher Scientific, 62248). Imagining was performed on the Leica DMi6000B TCS-SP5 II Inverted Confocal Microscope at ×1,000 magnification.

### Reporting summary

Further information on research design is available in the Nature Portfolio Reporting Summary linked to this article.

### Data availability

The NGS data from the PAM depletion assay and crRNA sequencing data were deposited at the NCBI GEO under accession code GSE178536. All other data supporting the findings in the Article and the Supplementary Information are available from the corresponding authors on reasonable request.

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

**Acknowledgements** We thank K. Makarova for guidance on naming Cas12a2 and for feedback on phylogenetic analyses; L. Fläxl for assistance with transcription–translation; L. Ostertag for assistance with nick translation; F. Ttofali for assistance with plasmid cloning and testing; D. Collias for suggestions around nicking translation; and A. Pawluk and A. Özcan for feedback on the manuscript. The plasmids pET28a-mH6-Cas12g1 (Addgene plasmid, 120879) and pACYC-Cas12g1 (Addgene plasmid, 120880) were gifts from Arbor Biosciences. Plasmid pC001 was a gift from F. Zhang (Addgene plasmid, 79150). This work was supported by an ERC Consolidator grant (865973 to C.L.B.), the DARPA Safe Genes program (HR0011-17-2-0042 to C.L.B.), the National Institutes of Health (R35GM138080 to R.N.J.), the Netherlands Organization for Scientific Research (NWO) through a Rubicon Grant (project 019.193EN.032, to I.M.), and the Federal Agency for Disruptive Innovation (to C.L.B.). The views, opinions and/or findings expressed should not be interpreted as representing the official views or policies of the Department of Defense or the US Government.

**Author contributions** Conceptualization: O.D., G.C.N., R.N.J. and C.L.B. Methodology: O.D., G.C.N., V.M.C., J.M., H.D., T.H. and D.J.K. Bioinformatic discovery of Cas12a2 orthologues: B.N.G. and O.D. First observation of bacterial abortive infection: G.C.N. Phylogenetic analysis: O.D. Discovery of RNA targeting: O.D., D.J.K., R.N.J. and C.L.B. Experimentation in *E. coli*: O.D., G.C.N., V.M.C., E.V., I.M. and J.W. Experimentation in transcription–translation: K.G.W., J.W. and

O.D. Experimentation in vitro: V.M.C., D.J.K., H.D, T.H. and J.M. Nick-translation diagnostic: T.G. Writing—original draft: O.D., V.M.C., R.N.J. and C.L.B. Writing—review and editing: all of the authors. Visualization: O.D., D.J.K., T.H., H.D., R.N.J. and C.L.B. Supervision: M.B.B., R.N.J. and C.L.B. Funding acquisition: R.N.J. and C.L.B.

**Funding** Open access funding provided by Helmholtz-Zentrum für Infektionsforschung GmbH (HZI).

**Competing interests** Benson Hill has one granted patent (US 9,896,696) and has filed additional patent applications. G.C.N. and M.B.B. are employees of Benson Hill. O.D., R.N.J. and C.L.B. have filed provisional patent applications on the related concepts, with one granted patent (US 9,896,696). C.L.B. is a co-founder of Locus Biosciences and is a scientific advisory board member of Benson Hill. The other authors declare no competing interests.

## Additional information
**Correspondence and requests for materials** should be addressed to Matthew B. Begemann, Ryan N. Jackson or Chase L. Beisel.

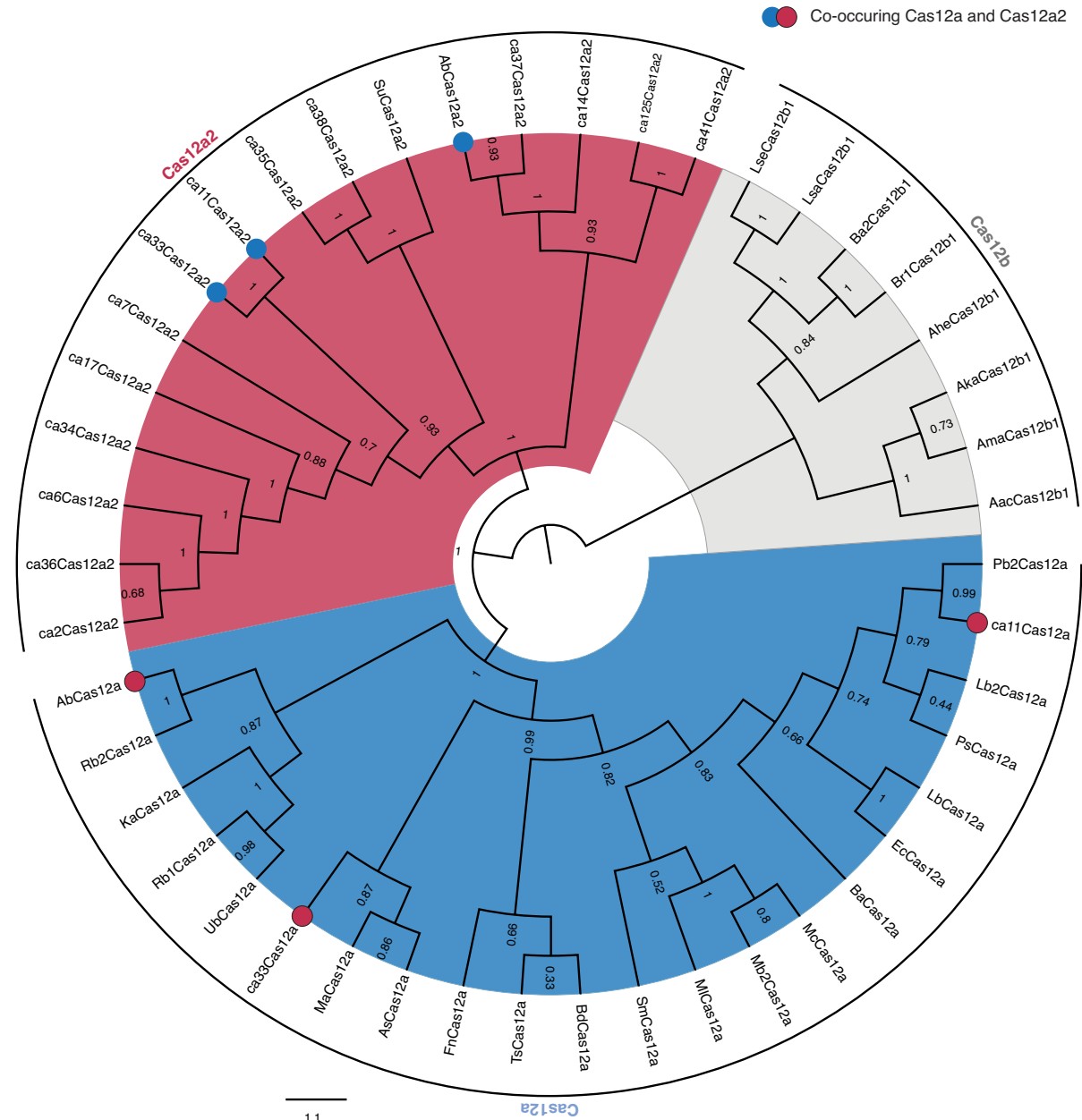

**Extended Data Fig. 1 | Maximum-likelihood phylogeny of Cas12a2 and Cas12a nucleases, with Cas12b as an outgroup.** The phylogenetic tree generated using amino acid sequences is the annotated version of Fig. 1a. The shaded regions designate Cas12a2 (red), Cas12a (blue), and Cas12b (grey) orthologues. The blue and red filled circles indicate systems that contain both Cas12a2 and Cas12a. Transfer bootstrap expectation (TBE) values are shown at the nodes.

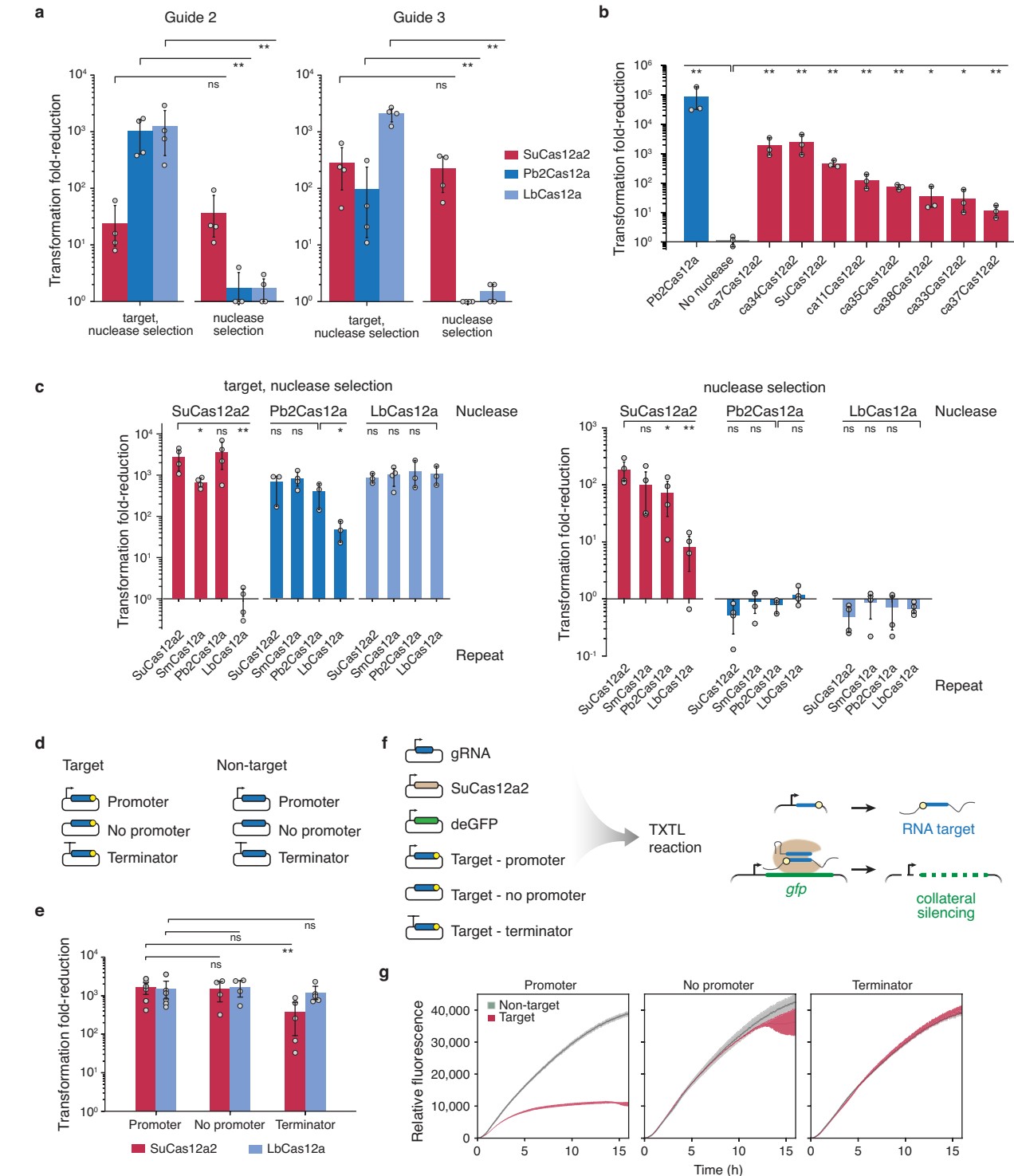

**Extended Data Fig. 2 | Cas12a2 can utilize Cas12a crRNA repeats and requires an RNA target in vivo. a**, Different guide:target pairs tested under antibiotic selection of the nuclease and target plasmids or only the nuclease plasmid. **b**, Reduction in plasmid transformation by Cas12a2 orthologues tested under antibiotic selection of the nuclease and target plasmids. **c**, Effect of swapping direct repeats associated with *Su*Cas12a2 and Cas12a nucleases on plasmid transformation. The indicated nuclease, repeat-encoding crRNA, and target were subjected to the traditional (left) and modified (right) plasmid interference assay in *E. coli*. **d**, Diagram of the target and non-target plasmids used in the plasmid clearance assay shown in e. **e**, Impact of a promoter and a terminator upstream of the target site on plasmid clearance by *Su*Cas12s2 and *Lb*Cas12a. **f**, Diagram of the cell-free TXTL reactions used to evaluate the impact of target expression on Cas12a2 collateral silencing shown in g. **g**, The effect of collateral silencing by *Su*Cas12a2 in TXTL as a function of having a promoter, no promoter, or terminator upstream of the target expression site. Scatter plots represent averages of 4 technical replicates ± s.d. Unless stated otherwise, values are means ± s.d. of at least 3 independent experiments started from separate colonies. ns: p > 0.05, *: p < 0.05, **: p < 0.005 calculated with one-tailed Welch's t-test.

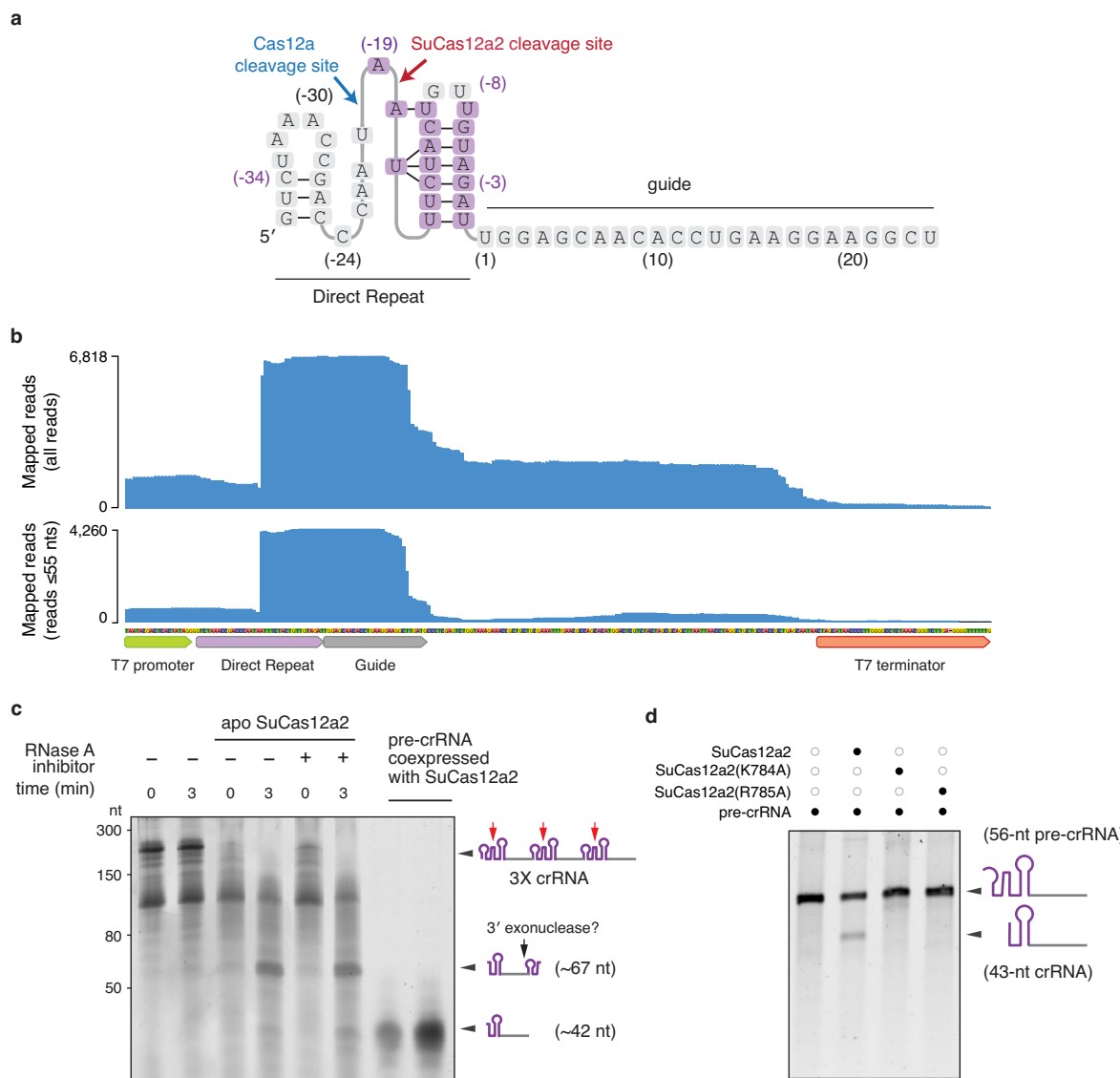

**Extended Data Fig. 3 | Pre-crRNA processing by *Su*Cas12a2. a**, Diagram of the predicted secondary structure of the *Su*Cas12a2 direct repeat. Cleavage sites of *Su*Cas12a2 and Cas12a are indicated. **b**, Sequencing coverage of the cDNA mapped to the crRNA locus in *E. coli* BL21(AI) expressing *Su*Cas12a2. Coverage for all of the quality-filtered reads above 10 nts as well as reads between 10 and 55 nts mapped to the plus strand are shown. **c**, In vitro processing of a 56-nt pre-crRNA incubated for 60 min in the presence of apo-*Su*Cas12a2 or two mutants predicted to disrupt crRNA processing. **d**, In vitro *Su*Cas12a2-mediated cleavage of a pre-crRNA containing three direct repeats and three spacers

(3× crRNA). Reactions containing apo-*Su*Cas12a2 are indicated. Time points after mixing the cleavage reaction with apo-*Su*Cas12a2 are indicated. The estimated sizes of the pre-crRNA and crRNAs after cleavage are indicated on the left. The last two lanes contain RNA extracted from *Su*Cas12a2 co-expressed with a CRISPR array. The difference in size between the major crRNA band in the in vitro assay (~ 67 nt) and the crRNA extracted from *Su*Cas12a2 bound to *E. coli*-expressed crRNA (~42) may be due to further trimming by 3′ exonucleases in the cell. For gel source data, see Supplementary Fig. 1.

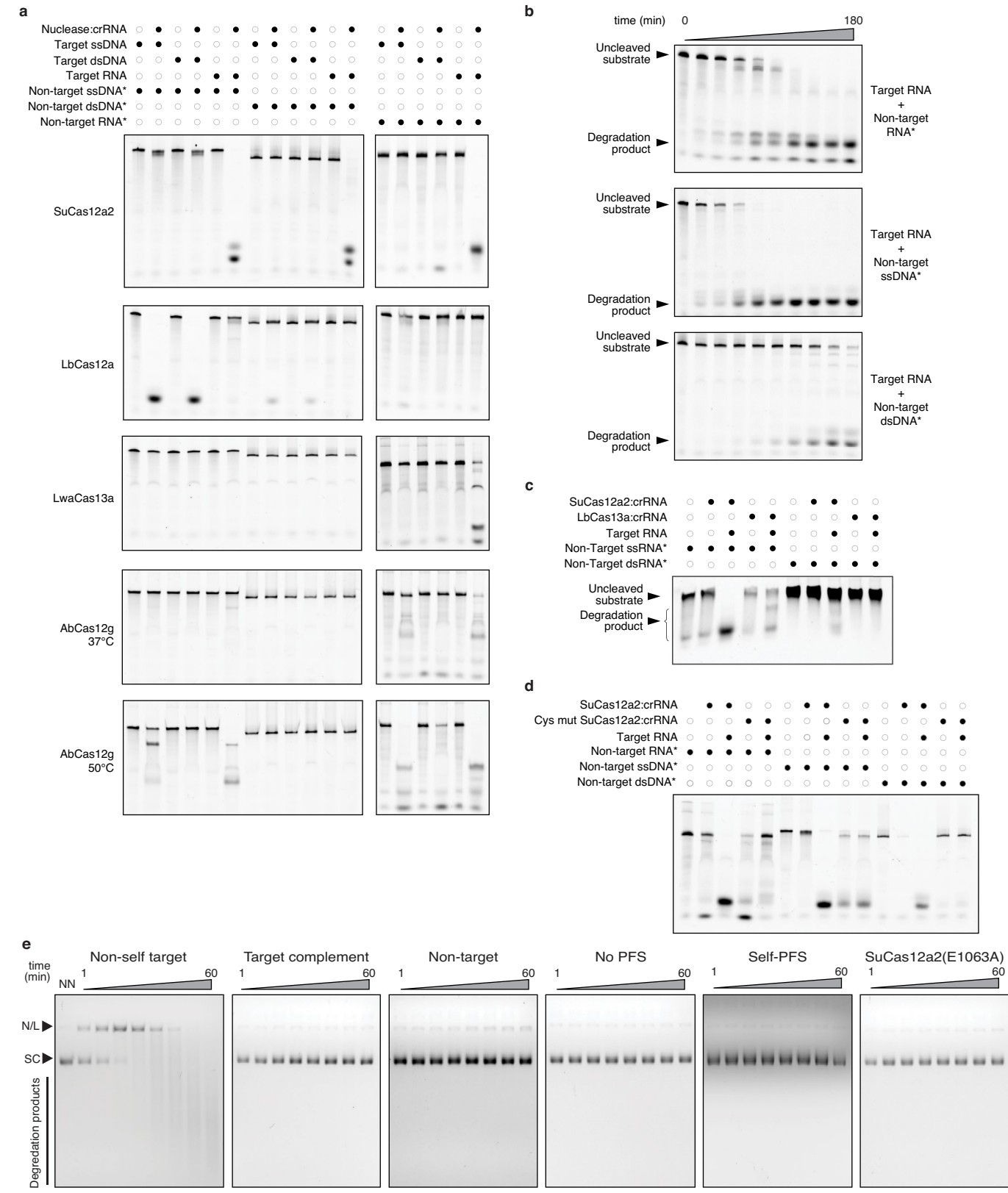

**Extended Data Fig. 4** | See next page for caption.

**Extended Data Fig. 4 | Properties of target recognition and collateral cleavage by *Su*Cas12a2 . a**, Non-specific collateral activities of *Su*Cas12a2, *Lb*Cas12a, *Lwa*Cas13a, and *Ab*Cas12g towards FAM-labelled non-target ssDNA, dsDNA, and RNA in the presence of target ssDNA, dsDNA, and RNA. All cleavage reactions were conducted at 37 °C unless specified otherwise. *Ab*Cas12g was triggered by RNA and ssDNA and exhibited preferential collateral cleavage of RNA over ssDNA but no discernable cleavage of dsDNA. **b**, *Su*Cas12a2-mediated cleavage over time of FAM-labelled RNA (top), ssDNA (middle), and dsDNA (bottom) non-target substrates. These substrates are cleaved by *Su*Cas12a2 through its non-specific collateral activity. **c**, Electromobility shift assay indicating *Su*Cas12a2 and *Lb*Cas13a mostly do not indiscriminately degrade dsRNA. Uncleaved substrates and cleaved products are indicated. **d**, Impact of mutating conserved cysteines within the predicted Zinc finger domain of *Su*Cas12a2 on RNA-triggered collateral activity. The mutated cysteines within *Su*Cas12a2 are C1170S, C1173S, C1188S and C1191S. **e**, Electromobility shift assays of supercoiled pUC19 plasmid with *Su*Cas12a2:crRNA and different RNA sequences over time. RNA sequences include: Non-self target (complementary to gRNA with 3′ GAAAG PFS), Target complement (complement of non-self target), Non-target (RNA sequence used in *trans*-cleavage assays), No PFS (RNA that only contains compliment to gRNA no 3′ PFS), Self-PFS (complementary to gRNA with 3′ AUCUA PFS). RNA sequences can be found in Supplementary Table 1. *Su*Cas12a2(E1063A) with the non-self target is included as a negative control. NN: no-nuclease control. For gel source data, see Supplementary Fig. 1.

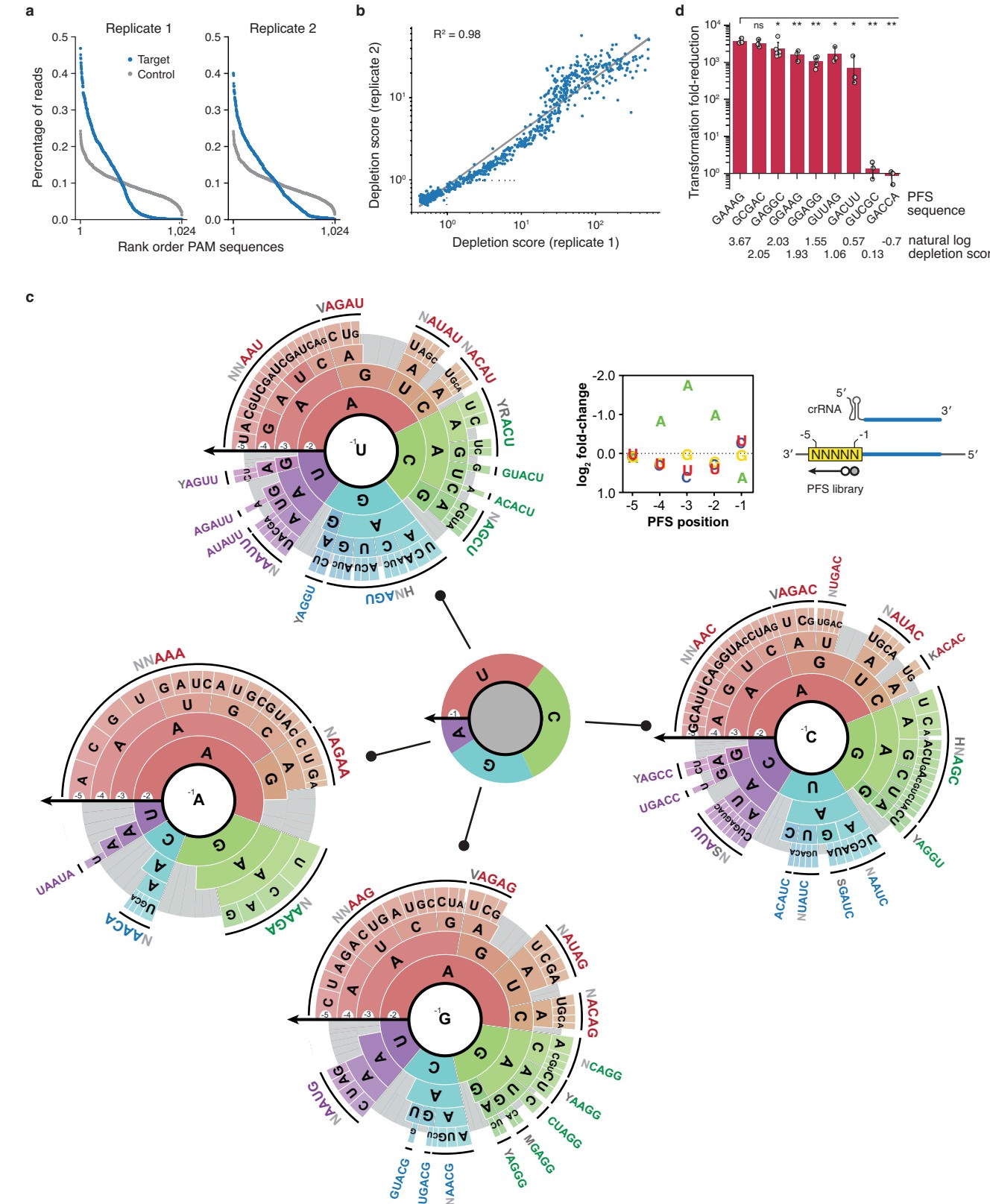

**Extended Data Fig. 5 | PFS depletion screen with *Su*Cas12a2 in *E. coli* BL21(AI).**
**a**, Sequencing coverage of the PFS libraries from the target and the control
*E. coli* cultures. Data from two biological replicates are shown. **b**, Correlation
between the depletion scores obtained from the two replicate libraries. **c**, The
complex PFS profile recognized by *Su*Cas12a2. See ref. [65] for more information
on interpreting PAM wheels. Given the complexity of the PFS profile, four

different PAM wheels are shown based on each nucleotide at the −1 PFS
position. **d**, Validation of selected PFS sequences identified in the PFS screen
with *Su*Cas12a2 using the plasmid clearance assay. Values are means ± s.d.
of ≥ 3 independent experiments started from separate colonies. ns: $p > 0.05$,
*: $p < 0.05$, **: $p < 0.005$ calculated with one-tailed Welch's t-test.

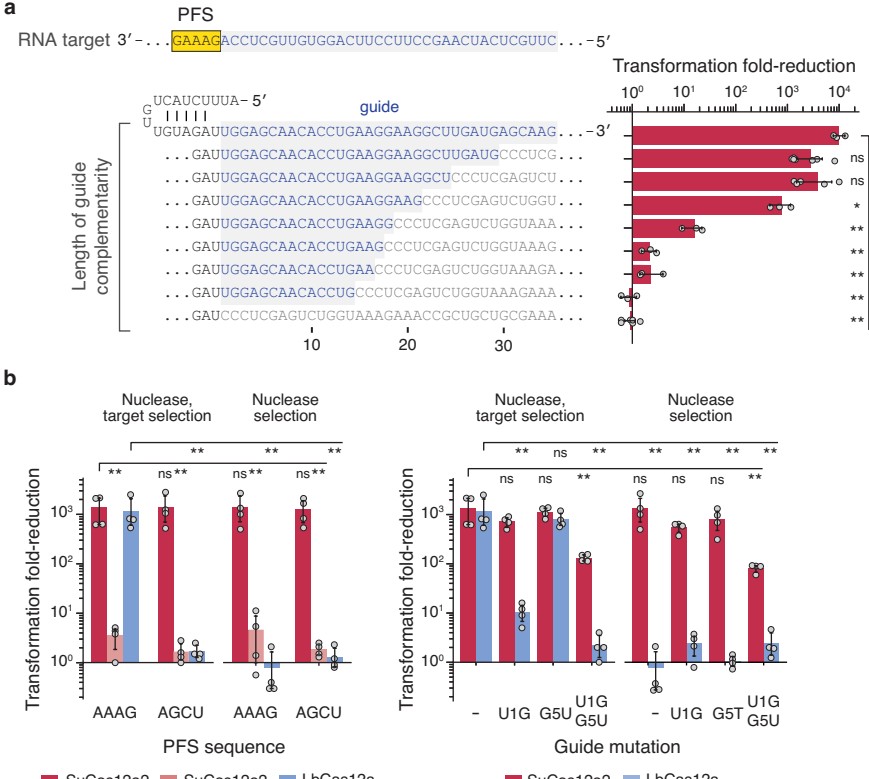

**Extended Data Fig. 6 | Impact of *Su*Cas12a2 guide and PFS alterations on plasmid targeting in *E. coli* BL21(AI). a**, Impact of guide length on plasmid clearance. The guide sequence is depicted with blue letters. The standard crRNA guide length based on crRNA processing (Extended Data Fig. 3) is 24 nts. **b**, Reduction in plasmid transformation by *Su*Cas12a2 and *Lb*Cas12a in the presence of a shared PAM/PFS, Cas12a2-specific PFS, and mismatches to the target under target plasmid and nuclease plasmid or nuclease plasmid-only selection. Values are means ± s.d. of at least 3 independent experiments started from separate colonies. ns: p > 0.05, *: p < 0.05, **: p < 0.005 calculated with one-tailed Welch's t-test.

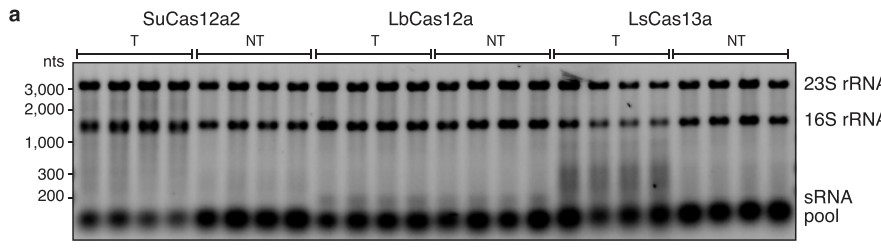

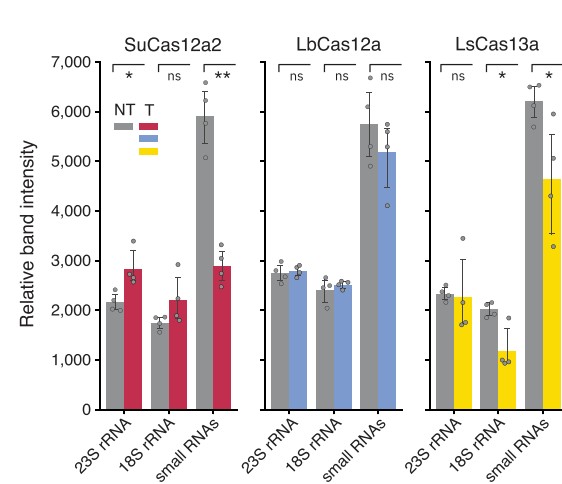

**Extended Data Fig. 7 | *Su*Cas12a2 degrades cellular RNA. a**, 1% agarose gel resolving total RNA extracted from *E. coli* BL21(AI) expressing *Su*Cas12a2, *Lb*Cas12a and *Ls*Cas13a under target (T) and non-target (NT) conditions. Expression of the nucleases and crRNA was induced with 10 nM IPTG and 0.2% arabinose for 2 h. Individual wells for each condition represent biological replicates. Nucleotide sizes are based on a resolved RNA ladder. sRNA pool: RNA pool comprising tRNAs and other small RNAs. **b**, Quantification of band intensities across the 4 independent experimental replicates started from separate colonies. Significance was calculated using two-tailed Welch's t-test (ns: p > 0.05, *: p < 0.05, **: p < 0.005).

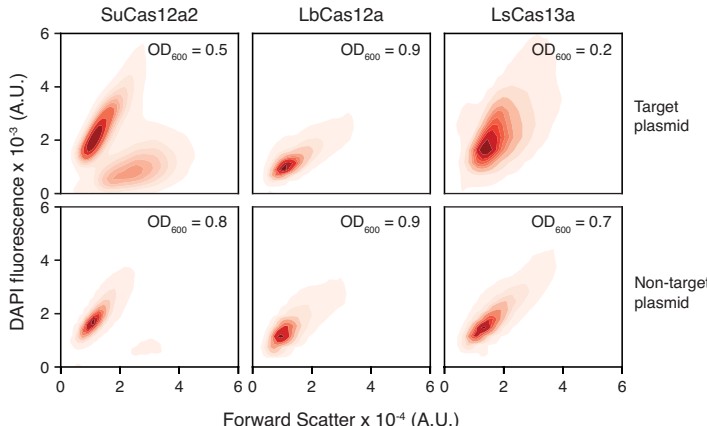

**Extended Data Fig. 8 | Flow cytometry analysis of the nuclease-expressing *E. coli* cells without antibiotic selection.** The cultures were collected 4 h after inoculation and induction of nuclease and crRNA. Prior to the analysis, the cells were stained with DAPI. The subpopulation with low DAPI and high forward scatter reflects elongated cells with reduced DNA content. Contour plots are representative of 4 independent experiments.

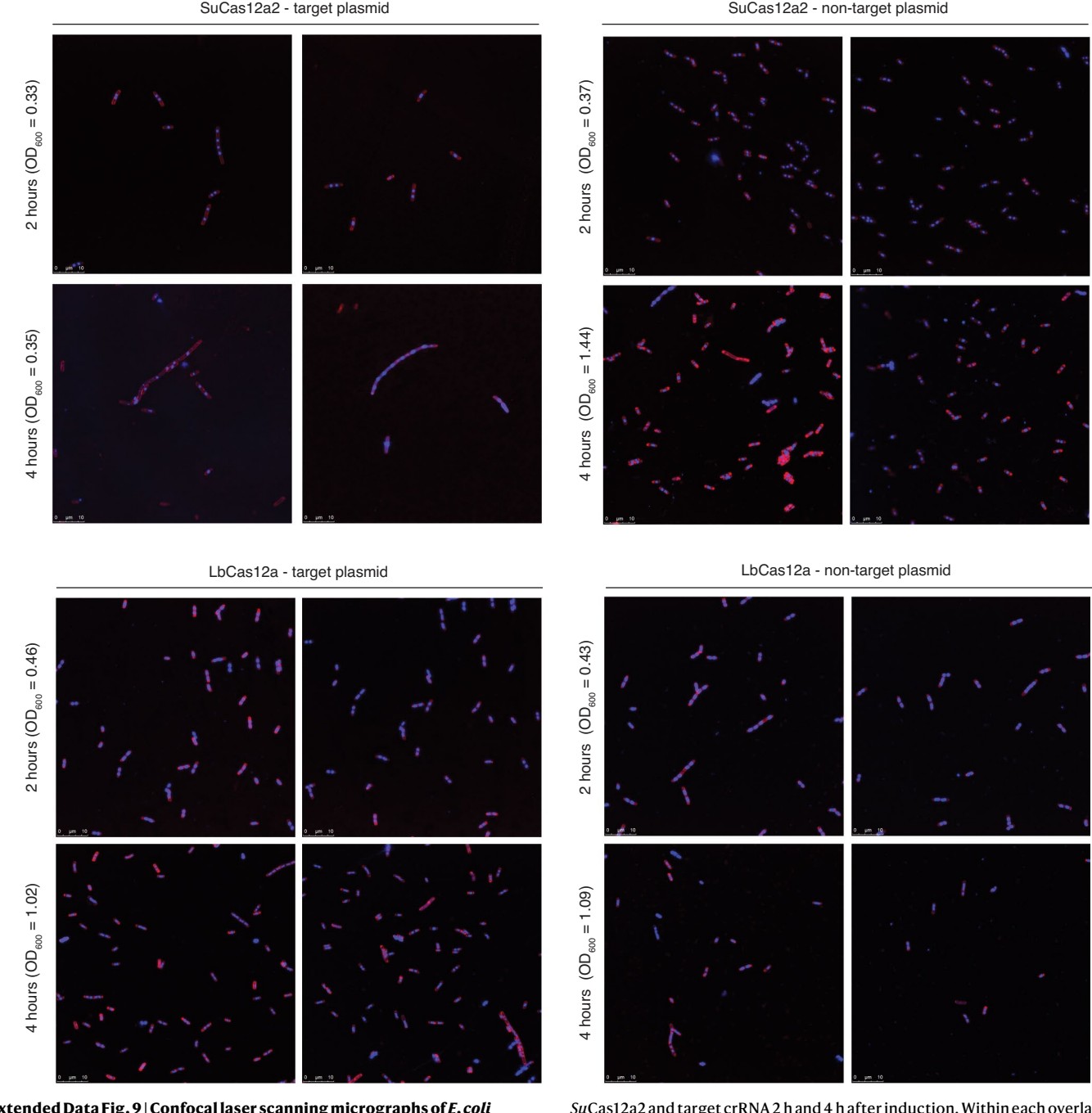

**Extended Data Fig. 9 | Confocal laser scanning micrographs of *E. coli* BL21(AI) transformed with *Su*Cas12a2–crRNA and *Lb*Cas12-crRNA expression plasmids under target and non-target conditions in the absence of antibiotic selection.** Extensive filamentation can be observed with the bacteria containing the target-expression plasmid and expressing *Su*Cas12a2 and target crRNA 2 h and 4 h after induction. Within each overlaid image, fluorescent DNA (DAPI) staining is shown in blue and membrane (FM4–64) staining is shown in red. Paired images are representative of 2 independent experiments.

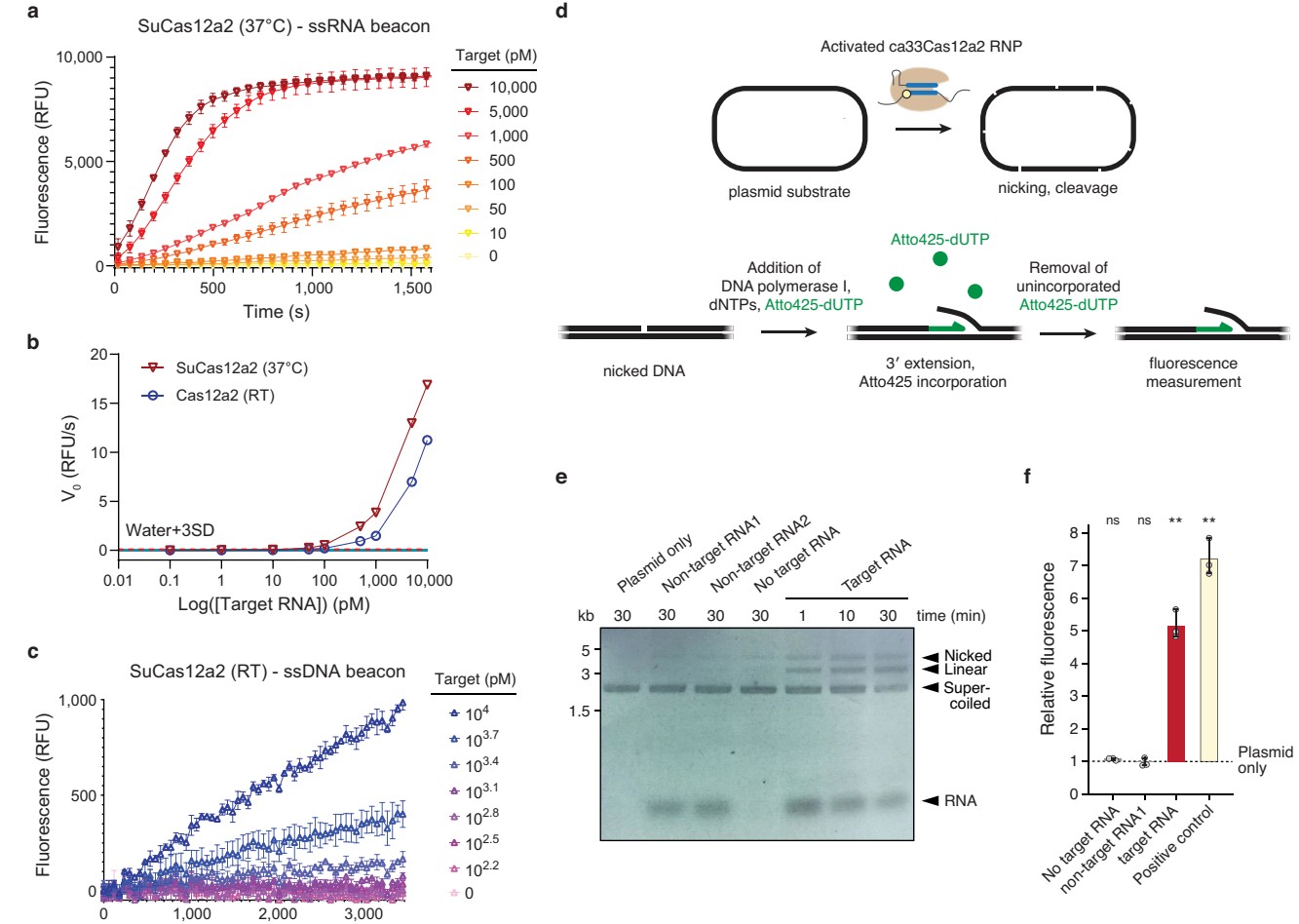

**Extended Data Fig. 10 | Assessment and extension of Cas12a2-based diagnostics. a**, Progress curve for target RNA-activated cleavage of RNA beacon by *Su*Cas12a2 at 37 °C. **b**, Limit of detection of *Su*Cas12a2 using an RNA beacon as determined using the velocity method[46]. Velocities were obtained by regression analysis of the linear regions of the progress curves. The velocity method was used to determine all reported limits of detection. **c**, Progress curve for target RNA-activated cleavage of ssDNA beacon by Cas12a2 at ambient temperature (RT). Error bars in a and c represent the mean and standard deviation of three independent measurements. Error bars in b represent the standard error for the linear fit of three independent measurements. **d**, Overview of detection assay based on plasmid nicking and nick translation. **e**, Resolved plasmid

substrate incubated with the ca33Cas12a2-gRNA RNP (100 nM) and the indicated RNA (1 nM). The incubation time before initiating nick translation is indicated for each experimental condition. The first lane (Plasmid only) represents the plasmid incubated without an RNP. Non-target RNA1: *COA1* gRNA and SARS-CoV-2 RNA. Non-target-RNA2: SARS-CoV-2 gRNA and COA1 RNA. **f**, Fluorescence measurements following nick translation with the different reactions from e. Fluorescence values were normalized to that of the plasmid-only control. For the positive control, the plasmid was subjected to a mix of DNase I and DNA polymerase I. Values are means ± s.d. of 3 independent experiments. Significance was calculated using two-tailed Welch's t-test compared to a value of 1 (ns: $p > 0.05$, *: $p < 0.05$, **: $p < 0.005$). For gel source data, see Supplementary Fig. 1.

| | |
|---|---|

# Reporting Summary

## Statistics

For all statistical analyses, confirm that the following items are present in the figure legend, table legend, main text, or Methods section.

| n/a | Confirmed | |
|---|---|---|
| ☐ | ☒ | The exact sample size (*n*) for each experimental group/condition, given as a discrete number and unit of measurement |
| ☐ | ☒ | A statement on whether measurements were taken from distinct samples or whether the same sample was measured repeatedly |
| ☐ | ☒ | The statistical test(s) used AND whether they are one- or two-sided *Only common tests should be described solely by name; describe more complex techniques in the Methods section.* |
| ☒ | ☐ | A description of all covariates tested |
| ☒ | ☐ | A description of any assumptions or corrections, such as tests of normality and adjustment for multiple comparisons |
| ☐ | ☒ | A full description of the statistical parameters including central tendency (e.g. means) or other basic estimates (e.g. regression coefficient) AND variation (e.g. standard deviation) or associated estimates of uncertainty (e.g. confidence intervals) |
| ☐ | ☒ | For null hypothesis testing, the test statistic (e.g. *F*, *t*, *r*) with confidence intervals, effect sizes, degrees of freedom and *P* value noted *Give P values as exact values whenever suitable.* |
| ☒ | ☐ | For Bayesian analysis, information on the choice of priors and Markov chain Monte Carlo settings |
| ☒ | ☐ | For hierarchical and complex designs, identification of the appropriate level for tests and full reporting of outcomes |
| ☒ | ☐ | Estimates of effect sizes (e.g. Cohen's *d*, Pearson's *r*), indicating how they were calculated |

*Our web collection on statistics for biologists contains articles on many of the points above.*

## Software and code

Policy information about availability of computer code

| | |
|---|---|
| Data collection | NovoExpress 1.5.6 |
| Data analysis | Python 3.9, pandas 1.2.0, matplotlib 3.5.2, Excel 16.62, NovoExpress 1.5.6, DBSCAN |

For manuscripts utilizing custom algorithms or software that are central to the research but not yet described in published literature, software must be made available to editors and reviewers. We strongly encourage code deposition in a community repository (e.g. GitHub). See the Nature Portfolio guidelines for submitting code & software for further information.

## Data

Policy information about availability of data

All manuscripts must include a data availability statement. This statement should provide the following information, where applicable:

- Accession codes, unique identifiers, or web links for publicly available datasets
- A description of any restrictions on data availability
- For clinical datasets or third party data, please ensure that the statement adheres to our policy

The NGS data from the PAM depletion assay and crRNA sequencing data were deposited to NCBI GEO under the accession GSE178536. All other data in the main text or the supplementary materials are available upon reasonable request.

## Human research participants

Policy information about studies involving human research participants and Sex and Gender in Research.

| | |
|---|---|
| Reporting on sex and gender | N/A |
| Population characteristics | N/A |
| Recruitment | N/A |
| Ethics oversight | N/A |

Note that full information on the approval of the study protocol must also be provided in the manuscript.

# Field-specific reporting

Please select the one below that is the best fit for your research. If you are not sure, read the appropriate sections before making your selection.

☒ Life sciences          ☐ Behavioural & social sciences          ☐ Ecological, evolutionary & environmental sciences

For a reference copy of the document with all sections, see nature.com/documents/nr-reporting-summary-flat.pdf

# Life sciences study design

All studies must disclose on these points even when the disclosure is negative.

| | |
|---|---|
| Sample size | At least three random colonies were picked as biological replication for each assay. The number of selected colonies were determined based on standards in the field for these types of experiments. |
| Data exclusions | No data were excluded from the analyses. |
| Replication | All of the measurements were performed in three, usually four, biological replicates, unless stated otherwise. |
| Randomization | Random colonies of transformants were picked as biological replicates. Other aspects of randomization (e.g., covariates) are not relevant. |
| Blinding | Blinding is not relevant to the study, as blinding is not normally included for bacterial and biochemical experiments. |

# Reporting for specific materials, systems and methods

We require information from authors about some types of materials, experimental systems and methods used in many studies. Here, indicate whether each material, system or method listed is relevant to your study. If you are not sure if a list item applies to your research, read the appropriate section before selecting a response.

## Materials & experimental systems

| n/a | Involved in the study |
|---|---|
| ☒ | Antibodies |
| ☒ | Eukaryotic cell lines |
| ☒ | Palaeontology and archaeology |
| ☒ | Animals and other organisms |
| ☒ | Clinical data |
| ☒ | Dual use research of concern |

## Methods

| n/a | Involved in the study |
|---|---|
| ☒ | ChIP-seq |
| ☐ | ☒ Flow cytometry |
| ☒ | MRI-based neuroimaging |

# Flow Cytometry

## Plots

Confirm that:

☒ The axis labels state the marker and fluorochrome used (e.g. CD4-FITC).

☒ The axis scales are clearly visible. Include numbers along axes only for bottom left plot of group (a 'group' is an analysis of identical markers).

☒ All plots are contour plots with outliers or pseudocolor plots.

☒ A numerical value for number of cells or percentage (with statistics) is provided.

## Methodology

| | |
|---|---|
| Sample preparation | For measuring DNA content, the cell pellets were resuspended in 1x PBS containing 2 µg/ml 4',6-diamidino-2-phenylindole (DAPI, ThermoFisher Scientific, 62248). The resuspended cells were stained for 10 minutes in the dark, after which 10 µl were transferred into 240 µl of 1x PBS on a 96 well plate. DAPI fluorescence was measured in the Pacific Blue spectrum (455 nm). Data regarding the forward scatter (FSC) and the side scatter (SSC) were also collected. For live/dead staining, the LIVE/DEAD BacLight Bacterial Viability and Counting Kit (Molecular Probes, L34856) kit was used. The samples were prepared following the kit's manual. Fluorescence was collected in the green (fluorescein for SYTO 9) and red (Texas Red for PI) channels. |
| Instrument | Cytoflow Novocyte Quanteon flow cytometer |
| Software | NovoExpress 1.5.6 |
| Cell population abundance | 50,000 total ungated events were measured per sample. Pure bacterial cell culture was used for the measurements. |
| Gating strategy | The dead cells in the live/dead staining experiment in each sample were gated based on the dead-cell suspension control treated with isopropyl alcohol. |

☒ Tick this box to confirm that a figure exemplifying the gating strategy is provided in the Supplementary Information.

