## [Peer Review File · Nature]

Manuscript Title: Cas12a2 elicits abortive infection via RNA-triggered destruction of dsDNA

Reviewer Comments & Author Rebuttals

Reviewer Reports on the Initial Version:

Referee expertise:

Referee #1: CRISPR biology

Referee #2: mechanisms of CRISPR immunity; structural biology

Referees' comments:

Referee #1:

The paper from Dmytrenko et al. characterizes SuCas12a2, an ortholog of the CRISPR-Cas type V Cas12a2 nuclease found in *Sulfuricurvum* sp. SuCas12a2 is an RNA-guided nuclease that can process its own guide RNAs and targets and cleaves complementary ssRNA targets. Subsequently, Cas12a2 displays collateral (sequence non-specific) cleavage activity on ssRNA, ssDNA, dsDNA and negatively-supercoiled DNA substrates, triggering an abortive infection (Abi) mechanism. The authors demonstrate that SuCas12a2-mediated targeting of complementary ssRNA substrates is dependent on a 5' PFS/PAM-like sequence. The authors demonstrate that many type V anti-CRISPRs were insufficient in blocking Cas12a2 due to structural differences (when compared to Cas12a). The authors conclude their study by showing that the versatile collateral activities of Cas12a2 can be harnessed for a sensitive RNA-detection/diagnostic tool.

This is a very thorough and comprehensive study, with an impressive set of experiments dissecting the many different catalytic activities (and phenotypic outcomes) of Cas12a2 and comparing them to other well-known nucleases such as Cas12a and Cas13a. The conclusions agree with the data presented, the experimental approaches are appropriate and the manuscript is well written. The authors present an interesting story highlighting the diversity in mechanisms of CRISPR Cas-mediated defense even when belonging to the same CRISPR-Cas type.

Major comments

The phylogenetic tree is highly surprising and very different from the latest tree from the Makarova and Koonin group. The tree includes several type V subtypes (Cas12a,b,c,d,f,g,j,k) and the TnpB ancestor. The Cas12a and Cas12a-2 end up at both sides of the dendrogram. If this were correct, Cas12a-2 should not be called a variant of Cas12a, but rather would be a representative of a new subtype that deserves to get a new name (such as Cas12n). The authors should comment on the approach that led to this unusual tree and why they used a name that suggests a strong phylogenetic relation with Cas12a.

Fig. 4c (and the corresponding text section): In contrast to most other parts of the manuscript, the description of these results is inadequate and deserves more attention to the different effects that can be seen. The authors appear to label the outcome as “less extensive RNA degradation” by comparing the breakdown of rRNA with that of Cas13a. This seems a bit arbitrary, as tRNA breakdown was more effective by Cas12a2 than Cas13a. The authors might also want to consider quantifying the results in Fig. 4c. More importantly, since the authors cannot exclude that RNA degradation contributes to the abortive phenotype they observe, the author should also reconsider the title of the manuscript.

It seems rather counterintuitive that targeting by Cas12a2 (and the resulting abortive infection-like phenotype) is more flexible (e.g. more tolerant to mismatches, flexible FPS, etc) when compared to Cas12a, when considering the impact on fitness of the host. One could argue that the decision to commit to altruistic cell death (Cas12a2) should be a more careful one than the decision to target a possible invading MGE (Cas12a). Can the authors comment on this?

Minor comments

Page 2, line 4: “However, a CRISPR-mediated abortive mechanism that relies on indiscriminate DNase activity has yet to be observed.”

Please revise; there are several type III auxiliary nucleases that act indiscriminately on DNA, such as NucC (Lau et al., Mol. Cell, 2020) and Can1 (McMahon et al., Nat. Commun., 2020). Whereas the authors are right that no experiments have been conducted to show that these enzymes produce an abortive infection-like phenotype, it has been shown for the recently characterized type III-associated Card1 protein (Rostøl et al., Nature, 2021). Also revise other instances, such as on p. 3, line 20: “...that in turn activate indiscriminate accessory RNases”. (type III immunity is not restricted to RNases.)

Page 2, line 10: “...stemming the infection”

Perhaps the authors meant to say “stalling” or “limiting” here?

Fig. 1f and Ext. Data Fig. 4

The authors tested multiple guides with similar outcomes. However, given the RNA-targeting nature of Cas12a2, these results would only make sense if these guides were 1) targeting the right strand and 2) were located in actively-transcribed regions on the plasmid. The authors should mention whether this was indeed the case. Later in the manuscript, the authors mention “that spurious transcription of our target plasmid is sufficient to activate the immune system in vivo”. Perhaps this hypothesis can be tested by targeting a self-splicing RNA (forming an RNA-only protospacer for Cas12a2)?

Some main and Ext. Data figures are not discussed in the right order. For example, on p. 5 from lines 86-92.

Page 6, line 93: “...an RNA-processing assay”

Please change to “...an in vitro pre-crRNA-processing assay”.

When discussing the pre-crRNA processing activity of Cas12a2, the 3' end trimming should be mentioned in the main text.

Page 6, line 117: "Of the three collateral substrates, ssRNA and ssDNA were more efficiently cleaved than ssDNA by Cas12a2"

Perhaps the authors meant to say "...than dsDNA by Cas12a2" here?

Page 7, line 124: "(called a protospacer flanking signal or PFS)"

PFS stands for protospacer flanking sequence.

Page 7, line 130: "Cas12a2 uses a "non-self activation" mechanism distinct from RNA-targeting systems that rely on tag:anti-tag "self deactivation" mechanisms"

It seems rather counterintuitive to draw this conclusion after presenting results showing the absence of collateral activity when using an tag:anti-tag RNA substrate (which would suggest the opposite).

Fig. 2e: Did the authors perform the appropriate control experiment to show that collateral plasmid DNA degradation is only induced in the presence of a target RNA (and not with a non-target RNA)?

From the results presented in Fig. 3 and Ext. Data Fig. 11 the authors conclude that Cas12a2 lacks a canonical seed. In Ext. Data Fig. 11 it can be seen that especially mutations starting at position ~20 have a dramatic impact on interference. Do these bases correspond with the "7-nts of pre-ordered crRNA [that] sit at the interface between REC1 and REC2 in conformation where bases are solvent exposed and primed for target recognition" as mentioned in the Bravo et al. manuscript? If so, the authors might want to investigate the impact of these (order) residues on interference.

Page 10, line 205: "To verify that loss of the nuclease- and crRNA-encoding plasmids does not contribute to this result"

This does not seem to make sense as 'this result' points to experiments where selection for the target plasmid was absent, not the nuclease (and crRNA) encoding plasmid.

Referee #2:

The authors report a distinct CRISPR-mediated abortive defense mechanism relying on non-specific nuclease activities of Cas12a2, which triggers the SOS damage and impairs cell growth by damaging the bacterial dsDNA. In addition to dsDNA, Cas12a2 also degrades ssRNA and ssDNA in a non-sequence-specific manner, upon binding of target RNA with a protospacer-flanking sequence. The authors have also demonstrated that Cas12a2 can be repurposed for RNA detection comparable to the existing RNA-detecting tools, which expands the CRISPR-based toolkit. This paper expands our knowledge of Cas12 family proteins and is therefore significant to the field.

Overall, the figures and text are clear, and the results are validated by appropriate controls. However, there are some specific issues that should be addressed.

1) Some figures and text location should be revised:

Line 107, "Similar to CRISPR-Cas systems that cause Abi, yet unlike the dsDNA-targeting Cas12a, Cas12a2 is activated only in the presence of complementary RNA targets."

Fig. 2a shows the Cas12a2 is activated in the presence of complementary RNA, but could not rule out the possibility that ssDNA or dsDNA also activate Cas12a2. For example, it could not rule out the possibility that binding of dsDNA activates Cas12a2, which degrades ssDNA or ssRNA. However, the data shown in Ext. Data Fig. 9a does.

Line 131: Thus, Cas12a2 uses a "non-self activation" mechanism distinct from RNA-targeting systems that rely on tag:anti-tag "self deactivation" mechanisms".

It seems that the self flank RNA contains a sequence, AUCUA, that is complementary to the 5'-repeat sequence, UAGAU. Based on the data in Fig. 2b, the authors could not get the conclusion that the Cas12a2 does not rely on tag: anti-tag "self deactivation". However, the data shown in Ext. Data Fig. 10d does.

2) Based on the non-self activation mechanism, could the authors explain the mechanisms of how Cas12a2 discriminate self and non-self RNA? Why could non-self RNA not activate Cas12a nuclease activities?

3) Given that Cas12a2-based RNA-detecting tool doesn't show advantage of the existing RNA-detecting tools. However, it would be interesting to develop the biotechnological tools based on the specific properties of Cas12a2, like degrading dsDNA and recognizing flexible PFS.

4) Line 118, "Of the three collateral substrates, ssRNA and ssDNA were more efficiently cleaved than dsDNA by Cas12a2.": Should the later ssDNA be dsDNA? If it is possible to measure the affinity of Cas12a2 to substrate (non-target RNA, non-target ssDNA and non-target dsDNA) and explain why Cas12a2 uses dsDNA to trigger SOS response on the condition that ssRNA and ssDNA were more efficiently cleaved than ssDNA.

5) In Fig. 2e, the plasmid DNA was fully cleaved in 60 min, but the data in Ext. Data Fig. 9b shows that the non-target dsDNA had not been fully cleaved in 180 min. Dose this mean the supercoiled dsDNA was cleaved much faster than linear dsDNA? Could the authors give some explanations for that?

6) Line 245: "Consistent with this observation, recent cryo-EM structures reveal Cas12a2 binds and cuts dsDNA with a mechanism completely distinct from all other CRISPR associated nucleases, while structure-guided mutants with impaired in vitro collateral dsDNase but not ssRNase and ssDNase activities abolished in vivo defense activity²⁰": The data in ref. 20 is not convincible with the conclusion that "structure-guided mutants with impaired in vitro collateral dsDNase but not ssRNase and ssDNase activities abolished in vivo defense activity." Could the authors explain why the Y465A mutant shows robust dsDNase activity but has low ability to clear plasmid? However, E1063A has completely lost dsDNA cleavage activity but has higher ability to clear plasmid than Y465A (Fig. 3h, I in ref. 20) . It would be better that the authors test these mutants mentioned in ref. 20.

Author Rebuttals to Initial Comments:

Manuscript 2022-06-09197

Rebuttal to the reviewers' comments

We thank both reviewers for taking the time to carefully review our manuscript and for their thoughtful and constructive feedback. Below we address each comment in blue. Related changes to the main text are in red. As part of these revisions, we made the following changes:

- Replaced the original phylogenetic trees with a more focused version demonstrating that Cas12a and Cas12a2 form distinct but related phylogenetic branches (Fig. 1a and Extended Data Fig. 1).
- Added plasmid clearance data demonstrating that a promoter at a target RNA site is not required to reduce plasmid transformation in *E. coli* with Cas12a2, but a terminator significantly decreases plasmid clearance (Extended Data Fig. 8b-c).
- Added cell-free transcription-translation (TXTL) data demonstrating that a promoter upstream of the target is necessary to drive activation of Cas12a2-mediated targeting (Extended Data Fig. 8d-e).
- Added several controls of supercoiled plasmid cleavage showing that a PFS-flanked target and a RuvC-intact Cas12a2 is required for cleavage (Extended Data Fig. 9e).
- Showed that target mutations that block immune activation by Cas12a are still recognized by Cas12a2 (Extended Data Fig. 10b).
- Quantified depletion of the small RNA pool when assessing collateral RNA cleavage in cells (Extended Data Fig. 15b).
- Devised a new diagnostic detection approach based on plasmid DNA and DNA nick translation that leverages the unique features of Cas12a2 (Extended Data Fig. 19d-f).

We also added examples of flow cytometry gating used as part of Figure 4b, which appears as Extended Data Fig. 14.

We believe these changes further strengthen our scientific claims and the overall impact of the story.

Referee #1:

The paper from Dmytrenko et al. characterizes SuCas12a2, an ortholog of the CRISPR-Cas type V Cas12a2 nuclease found in *Sulfuricurvum* sp. SuCas12a2 is an RNA-guided nuclease that can process its own guide RNAs and targets and cleaves complementary ssRNA targets. Subsequently, Cas12a2 displays collateral (sequence non-specific) cleavage activity on ssRNA, ssDNA, dsDNA and negatively-supercoiled DNA substrates, triggering an abortive infection (Abi) mechanism. The authors demonstrate that SuCas12a2-mediated targeting of complementary ssRNA substrates is dependent on a 5' PFS/PAM-like sequence. The authors demonstrate that many type V anti-CRISPRs were insufficient in blocking Cas12a2 due to structural differences (when compared to Cas12a). The authors conclude their study by showing

that the versatile collateral activities of Cas12a2 can be harnessed for a sensitive RNA-detection/diagnostic tool.

This is a very thorough and comprehensive study, with an impressive set of experiments dissecting the many different catalytic activities (and phenotypic outcomes) of Cas12a2 and comparing them to other well-known nucleases such as Cas12a and Cas13a. The conclusions agree with the data presented, the experimental approaches are appropriate and the manuscript is well written. The authors present an interesting story highlighting the diversity in mechanisms of CRISPR Cas-mediated defense even when belonging to the same CRISPR-Cas type.

We thank the reviewer for their supportive summary of our work as well as their helpful feedback on how to improve the manuscript.

Major comments

The phylogenetic tree is highly surprising and very different from the latest tree from the Makarova and Koonin group. The tree includes several type V subtypes (Cas12a,b,c,d,f,g,j,k) and the TnpB ancestor. The Cas12a and Cas12a-2 end up at both sides of the dendrogram. If this were correct, Cas12a-2 should not be called a variant of Cas12a, but rather would be a representative of a new subtype that deserves to get a new name (such as Cas12n). The authors should comment on the approach that led to this unusual tree and why they used a name that suggests a strong phylogenetic relation with Cas12a.

We reached out to Dr. Makarova about this discrepancy, and she noted that they do not use phylogenetic reconstructions between the subtypes because of the low number of phylogenetically important positions present across Cas12 nucleases. Instead, they utilize similarity dendrograms that involve distinct methodology for clustering related sequences but without modeling phylogenetic assumptions. Under this framework, their analyses led them to classify Cas12a2 nucleases as variants of Cas12a.

Rather than replicate the extensive analyses performed by Koonin and Makarova, in response to the reviewer's concern, we instead focused our analysis on comparing our set of nucleases with Cas12a nucleases. The resulting analysis (see below) shows that Cas12a (blue) and Cas12a2 (red) nucleases form monophyletic clades that share a common ancestor when compared to a Cas12b outgroup. This type of analysis is in line with other papers that have reported the diversity of Cas nucleases within a subtype (e.g., Chen *et al.*, *Front Microbiol* 2019; Goltsman *et al.*, *CRISPR J* 2020), including prior work from Koonin and Makarova (e.g., Makarova *et al.*, *CRISPR J* 2018; Makarova *et al.*, *Nat Rev Microbiol* 2020).

These analyses are conveyed as new versions of Figure 1a and Extended Data Figure 1. We also revised the beginning of the results section to better reflect the consistency between the prior work by Koonin and Makarova and our own work with the following.

Page 4: “Cas12a2 comprises a group of type V effector nucleases related to Cas12a¹⁶, with Cas12a2 orthologs previously being classified as Cas12a variants¹⁹. Our analyses similarly place them in a monophyletic clade that shares the last common ancestor with Cas12a nucleases (**Fig. 1a and Extended Data Fig. 1**).”

Given our reliance on the prior work by Koonin and Makarova, we opted to retain the name “Cas12a2”, which indicates it is a variant of Cas12a. The CRISPR field may eventually deem Cas12a2 a distinct subtype of Type V CRISPR nucleases based on our findings, where we expect such decisions to be made by the community through the next update to the last classification scheme for CRISPR-Cas systems (Makarova et al. *Nat Rev Microbiol* 2020).

Fig. 4c (and the corresponding text section): In contrast to most other parts of the manuscript, the description of these results is inadequate and deserves more attention to the different effects that can be seen. The authors appear to label the outcome as “less extensive RNA degradation” by comparing the breakdown of rRNA with that of Cas13a. This seems a bit arbitrary, as tRNA breakdown was more effective by Cas12a2 than Cas13a. The authors might also want to consider quantifying the results in Fig. 4c. More importantly, since the authors cannot exclude that RNA degradation contributes to the abortive phenotype they observe, the author should also reconsider the title of the manuscript.

The reviewer raised an excellent point. We similarly recognize the apparent degradation of small RNAs including tRNAs. Following the above suggestion, we quantified the bands corresponding to 23S rRNA, 18S rRNA, and small RNAs and confirmed significant ($p < 0.005$) reduction in the intensity of the small RNA bands under SuCas12a2-targeting conditions. This analysis complements the overall RNA degradation visible on the gel under the targeting condition with SuCas12a2 and LsCas13a. The resulting figure shown below is included as Extended Data Figure 15b. We also added the following:

Page 11: “While Cas13a significantly depleted both rRNAs and the small RNA pool that includes tRNAs, Cas12a2 significantly depleted only the small RNA pool (**Fig. 4c and Extended Data Fig. 15**).”

The reviewer also rightfully pointed out that RNase activity may contribute to abortive infection, particularly given the depletion of small RNAs. While the clamp mutants in Bravo *et al.* demonstrate that the dsDNase activity is required for plasmid interference, the RNase activity may still play a role for other invaders. Therefore, we made the multiple changes in the manuscript to avoid implying that DNase activity is the sole activity that can drive interference. These changes included the following examples:

Page 2: "...a CRISPR-mediated abortive mechanism that leverages indiscriminate DNase activity of a single-effector nuclease has yet to be observed."

Page 8: "...the RNA-targeting ssRNase, ssDNase, and dsDNase are unique to SuCas12a2 (**Extended Data Fig. 9a**). Collectively, these *in vitro* results reveal that crRNAs guide SuCas12a2 to RNA targets, activating RuvC-dependent cleavage of ssRNA, ssDNA, and dsDNA. These activities, in part, or in total, potentially underlie the Abi phenotype."

After carefully considering feedback from the reviewer, we decided to retain the original manuscript title, as it does not rule out RNase activity also contributing to immune defense but points out the distinct immunological property of this single-effector nuclease.

It seems rather counterintuitive that targeting by Cas12a2 (and the resulting abortive infection-like phenotype) is more flexible (e.g. more tolerant to mismatches, flexible FPS, etc) when compared to Cas12a, when considering the impact on fitness of the host. One could argue that the decision to commit to altruistic cell death (Cas12a2) should be a more careful one than the decision to target a possible invading MGE (Cas12a). Can the authors comment on this?

We can see the reviewer's point that the flexibility of Cas12a2 combined with its detrimental impact on the host is counterintuitive when compared to Cas12a. This oddity may boil down to the defense strategy of either targeting DNA or RNA. In the case of DNA phages, Cas12a would act first on the DNA, clearing the invader and sparing the host before any phage transcripts can

accumulate. However, if Cas12a cannot effectively cleave the DNA (e.g., mutated target/PAM, DNA modifications), then Cas12a2 could be activated to sacrifice the host and prevent further spread of the invader. We believe this could help explain the difference in flexibility for the dual-nuclease systems. As direct support, we explicitly tested target mutations, showing that these mutations would disable immunity by Cas12a but not Cas12a2. The resulting data are shown below and were incorporated as Extended Data Figure 10b.

We also added the following to capture these new data:

Page 10: “Additionally, the promiscuity further allowed SuCas12a2 to recognize target mutations that disrupt targeting by Cas12a (**Extended Data Fig. 10b**).”

Even for CRISPR-Cas systems possessing only Cas12a2, the flexibility parallels other RNA-targeting CRISPR-Cas systems. For instance, Cas13 and the Cmr complex normally rely on tag:anti-tag complementarity that can accommodate many more sequences than the stringency provided by typical PAM recognition in DNA targeting systems. For Cas12a2, the flexible PFS is still more stringent than tag:anti-tag complementarity, which in some cases has been described as the absence of a single nucleotide (Abudayyeh *et al.*, Science 2016); flexibility in guide:target mismatches could help compensate for the more stringent PFS. Finally, as we only thoroughly characterized SuCas12a2, other orthologs may exhibit different stringencies.

To address this point we have added the following text to the discussion:

Page 13: “This flexibility in target recognition mirrors the flexibility of tag:anti-tag complementarity observed with Types III and VI systems^{3,4} and could be particularly advantageous against rapidly evolving phages, although other Cas12a2 orthologs must be characterized to determine whether such promiscuity is a common feature of these nucleases. The flexibility could further serve as a back-up mechanism if precise recognition and clearance of DNA targets by Cas12a fails in organisms that encode both Cas12a and Cas12a2 adjacent to a single CRISPR array (**Fig. 1a,d and Extended Data Figures 1 and 11b**).”

Minor comments

Page 2, line 4: “However, a CRISPR-mediated abortive mechanism that relies on indiscriminate DNase activity has yet to be observed.”

Please revise; there are several type III auxiliary nucleases that act indiscriminately on DNA, such as NucC (Lau et al., Mol. Cell, 2020) and Can1 (McMahon et al., Nat. Commun., 2020). Whereas the authors are right that no experiments have been conducted to show that these enzymes produce an abortive infection-like phenotype, it has been shown for the recently characterized type III-associated Card1 protein (Rostøl et al., Nature, 2021). Also revise other instances, such as on p. 3, line 20: “...that in turn activate indiscriminate accessory RNases”. (type III immunity is not restricted to RNases.)

The reviewer rightfully notes that Card1, a ssDNase activated in *trans* by triggered Type III systems, was shown to drive dormancy in Rostøl et al., 2021. To better reflect this prior work, we revisited all of our descriptions of how CRISPR-Cas systems have been shown to induce an abortive infection and updated the text to make sure Card1 is reflected. For instance, we made the following changes:

Page 2: “Several RNA-targeting CRISPR-Cas systems (e.g., types III and VI) cause Abi phenotypes by activating indiscriminate nucleases. However, a CRISPR-mediated abortive mechanism that leverages indiscriminate DNase activity through a single-effector nuclease is yet to be observed.”

Page 3: “In type III systems, target RNA binding triggers production of cyclic oligoadenylate secondary messengers that in turn activate indiscriminate accessory RNases and single-stranded (ss)DNases that can drive Abi^{4,5,9-11}. Additionally, it has been proposed that Abi is mediated by indiscriminate dsDNases (e.g. NucC) activated by type III secondary messengers or indiscriminate ssDNase activity from type V Cas12a single-effector nucleases¹⁴. However, type III CRISPR-mediated dsDNase activity has yet to be examined in vivo, and the ssDNase activity of Cas12a was recently shown to not cause Abi¹⁵.”

Page 5: “CRISPR systems that cause Abi phenotypes (e.g., types III and VI) rely on indiscriminate nucleases activated by RNA targeting.”

Page 2, line 10: "...stemming the infection"

Perhaps the authors meant to say "stalling" or "limiting" here?

We agree that a better word could be used here. We replaced "stemming" with "thwarting".

Fig. 1f and Ext. Data Fig. 4

The authors tested multiple guides with similar outcomes. However, given the RNA-targeting nature of Cas12a2, these results would only make sense if these guides were 1) targeting the right strand and 2) were located in actively-transcribed regions on the plasmid. The authors should mention whether this was indeed the case. Later in the manuscript, the authors mention "that spurious transcription of our target plasmid is sufficient to activate the immune system in vivo". Perhaps this hypothesis can be tested by targeting a self-splicing RNA (forming an RNA-only protospacer for Cas12a2)?

We agree with the reviewer that these points need to be clarified and possibly confirmed for our *in vivo* work. When testing different target sites, we now note with the following that the targets were cloned in the same location and orientation on the plasmid because RNA targeting was not apparent at that point in the story:

Page 5: "Similar trends were observed with different targets cloned into the same plasmid location".

Following the reviewer's comment, we specifically explored the role of spurious transcription given that our original target constructs did not explicitly encode an upstream promoter. In support of spurious transcription, introducing a terminator upstream of the target significantly reduced plasmid interference in *E. coli*, whereas an upstream promoter was required to detect collateral activity of SuCas12a2 in a cell-free transcription-translation assay. The associated data (see below) were incorporated into Extended Data Figure 8.

We also added the following to account for the new insights.

Page 6: “The potency of plasmid interference with SuCas12a2 (Fig. 1f) was curious given the lack of a defined promoter upstream of the target in this construct. However, we attribute interference to spurious transcription of the encoding plasmid for two reasons: introducing an upstream terminator significantly reduced plasmid interference in *E. coli* (Extended Data Fig. 8b-c), and an upstream promoter was required to detect collateral activity in a cell-free transcription-translation assay²³ (Extended Data Fig. 8d-e).”

Some main and Ext. Data figures are not discussed in the right order. For example, on p. 5 from lines 86-92.

As suggested by the reviewer, we carefully checked the numbering and order of the Extended Data Figures. Any errors were corrected--particularly swapping Extended Data Figures 2 and 3.

Page 6, line 93: “...an RNA-processing assay”
 Please change to “...an in vitro pre-crRNA-processing assay”.

In line with the reviewer’s suggestion, we revised the sentence to read the following.

Page 6: “Consistent with this prediction, RNA sequencing of pre-crRNAs processed by SuCas12a2 *in vitro* revealed that processing occurs one nucleotide (nt) downstream of the position cleaved by Cas12a (Extended Data Fig. 7a).”

When discussing the pre-crRNA processing activity of Cas12a2, the 3' end trimming should be mentioned in the main text.

We agree that 3' trimming should be mentioned as part of crRNA maturation. We therefore added the following.

Page 6: “The 3' end of the spacer also underwent trimming to form an ~24-nt guide (**Extended Data Fig. 7c**), possibly through host ribonucleases as observed for Cas9 crRNAs.”

Page 6, line 117: “Of the three collateral substrates, ssRNA and ssDNA were more efficiently cleaved than ssDNA by Cas12a2”

Perhaps the authors meant to say “...than dsDNA by Cas12a2” here?

We thank the reviewer for pointing out this unfortunate typo. ssDNA was replaced with dsDNA in the text as suggested by the reviewer.

Page 7, line 124: “(called a protospacer flanking signal or PFS)”

PFS stands for protospacer flanking sequence.

We thank the reviewer for catching this incorrect definition. We have updated the text as suggested.

Page 7, line 130: “Cas12a2 uses a “non-self activation” mechanism distinct from RNA-targeting systems that rely on tag:anti-tag “self deactivation” mechanisms”

It seems rather counterintuitive to draw this conclusion after presenting results showing the absence of collateral activity when using an tag:anti-tag RNA substrate (which would suggest the opposite).

We thank the reviewer for raising this point. The key result from which we drew our conclusion was that eliminating the flanking PFS sequence altogether prevented targeting. In revisiting the sentence, we decided the most reasonable approach was to delete this sentence.

Fig. 2e: Did the authors perform the appropriate control experiment to show that collateral plasmid DNA degradation is only induced in the presence of a target RNA (and not with a non-target RNA)?

We thank the reviewer for pointing out the lack of appropriate controls with this experiment. To address this concern, we assessed plasmid targeting *in vitro* with a non-cognate target or no PFS or a self-PFS as well as with the RuvC mutant. None of these controls demonstrated plasmid

cleavage. The resulting data (see below) were incorporated as Extended Data Figure 9e, and we added the following.

Page 8: “We observed that SuCas12a2 rapidly nicked, linearized, and degraded pUC19 DNA (Fig. 2e), but only in the presence of a cognate target and PFS and with an intact RuvC domain (Extended Data Fig. 9e).”

From the results presented in Fig. 3 and Ext. Data Fig. 11 the authors conclude that Cas12a2 lacks a canonical seed. In Ext. Data Fig. 11 it can be seen that especially mutations starting at position ~20 have a dramatic impact on interference. Do these bases correspond with the “7-nts of pre-ordered crRNA [that] sit at the interface between REC1 and REC2 in conformation where bases are solvent exposed and primed for target recognition” as mentioned in the Bravo et al. manuscript? If so, the authors might want to investigate the impact of these (ordered) residues on interference.

We agree with the reviewer that the data presented in Extended Data Fig. 11 demonstrates that consecutive mismatches at the 3' end of the guide disrupts nuclease activity, especially once only 14 nts are available to base pair with the target. Consistent with this observation, in the accompanying manuscript, RNA targets with 10 nts removed from the 5' end do not activate cleavage. We agree with the reviewer that the accompanying manuscript suggests the pre-ordered RNA in the binary structure may serve as a type of “seed” for initiating base pairing; however, we also believe that our mutational analysis supports our statement that such a “seed” is non-canonical in that up to two mismatches are tolerated in the last 7 nts of the guide RNA. Additionally, because the density of the crRNA is non-continuous in the binary structure, we cannot assign exact identity to the bases that are pre-ordered.

To clarify how our *in vivo* data fit with the structural observation of a pre-ordered segment of the crRNA guide, we have added the following.

Pages 9-10: “The flexible PFS recognition and a tolerance for guide:target mismatches indicate that SuCas12a2 exhibits promiscuous target recognition and appears to lack a canonical seed hypersensitive to guide-target mismatches. However, the necessary pairing

with the 3' end of the guide is consistent with this end being pre-ordered in the structure of the crRNA:Cas12a2 binary complex and possibly initiating base pairing with the target.”

Page 10, line 205: “To verify that loss of the nuclease- and crRNA-encoding plasmids does not contribute to this result”

This does not seem to make sense as ‘this result’ points to experiments where selection for the target plasmid was absent, not the nuclease (and crRNA) encoding plasmid.

We thank the reviewer for pointing out insufficient clarity in the mentioned sentence. Here we addressed a scenario in which triggered SuCas12a2 could clear the nuclease plasmid in the presence of the target plasmid, which would impact transformation because both plasmids were under antibiotic selection. To better clarify this scenario in the text, we updated the sentence to read the following:

Page 10: “Reduced plasmid transformation could have resulted from clearance of the nuclease- and crRNA-encoding plasmid under antibiotic selection. Therefore, we evaluated growth of *E. coli* in liquid culture following induction of SuCas12a2, LbCas12a, or LsCas13a with a targeting crRNA under different antibiotic selection conditions including a no-antibiotic condition (**Fig. 4a**). Under these conditions, both SuCas12a2 and LsCas13a suppressed culture growth even in the absence of plasmid selection, while LbCas12a only suppressed growth in the presence of the target plasmid antibiotic.”

We tested all possible conditions, including the selection for the nuclease encoding plasmid only and no-antibiotic selection, as shown in Figure 4a.

Referee #2:

The authors report a distinct CRISPR-mediated abortive defense mechanism relying on non-specific nuclease activities of Cas12a2, which triggers the SOS damage and impairs cell growth by damaging the bacterial dsDNA. In addition to dsDNA, Cas12a2 also degrades ssRNA and ssDNA in a non-sequence-specific manner, upon binding of target RNA with a protospacer-flanking sequence. The authors have also demonstrated that Cas12a2 can be repurposed for RNA detection comparable to the existing RNA-detecting tools, which expands the CRISPR-based toolkit. This paper expands our knowledge of Cas12 family proteins and is therefore significant to the field.

Overall, the figures and text are clear, and the results are validated by appropriate controls. However, there are some specific issues that should be addressed.

We thank the reviewer for their summary and support of our work. We address the specific issues raised by the reviewer below.

1) Some figures and text location should be revised:

Line 107, “Similar to CRISPR-Cas systems that cause Abi, yet unlike the dsDNA-targeting Cas12a, Cas12a2 is activated only in the presence of complementary RNA targets.”

Fig. 2a shows the Cas12a2 is activated in the presence of complementary RNA, but could not rule out the possibility that ssDNA or dsDNA also activate Cas12a2. For example, it could not rule out the possibility that binding of dsDNA activates Cas12a2, which degrades ssDNA or ssRNA. However, the data shown in Ext. Data Fig. 9a does.

Line 131: Thus, Cas12a2 uses a “non-self activation” mechanism distinct from RNA-targeting systems that rely on tag:anti-tag “self deactivation” mechanisms”.

It seems that the self flank RNA contains a sequence, AUCUA, that is complementary to the 5'-repeat sequence, UAGAU. Based on the data in Fig. 2b, the authors could not get the conclusion that the Cas12a2 does not rely on tag: anti-tag “self deactivation”. However, the data shown in Ext. Data Fig. 10d does.

We thank the reviewer for noting inconsistencies they saw between these claims and the cited figures. In the case of line 107, the data in Figure 2a show that the RNA target but not the ssDNA or dsDNA targets lead to cleavage activity. Extended Data Figure 9a provides a more complete dataset but also explores collateral activity that had not been revealed yet. Given this, we believe Figure 2a sufficiently supports the claim without needing to cite Extended Data Fig. 9a.

In the case of line 131, we agree with the reviewer that the claim is not fully supported based on Figure 2b. Considering that we provide data refuting the tag:anti-tag interaction as part of Extended Data Figure 10d, we decided to remove the sentence noted by the reviewer.

2) Based on the non-self activation mechanism, could the authors explain the mechanisms of how Cas12a2 discriminate self and non-self RNA? Why could non-self RNA not activate Cas12a nuclease activities?

The reviewer poses a great question about self/non-self recognition by Cas12a2. Self recognition would arise from antisense transcription of the CRISPR array. However, the 3' end of the encoding repeat strongly diverges from the set of recognized PFS sequences, preventing Cas12a2 activation despite perfect complementarity with the crRNA guide. This mechanism parallels DNA targeting CRISPR-Cas systems, where the PAM-containing portion of the repeat strongly diverges from the recognized PAM to prevent self recognition. To capture these insights, we added the following to the discussion:

Page 13: "While flexible, the PFS would still prevent self-recognition of spurious antisense transcription of the CRISPR array, as the corresponding PFS-containing portion of the repeat strongly diverges from the recognized PFS--a standard feature of self/nonself-recognition for PAMs in DNA-targeting CRISPR-Cas systems⁴⁸."

The reviewer also asked why non-self RNA would not activate Cas12a. This scenario can be discarded because Cas12a cannot recognize RNA targets as established in the literature as well as with our own data (Extended Data Fig. 9).

3) Given that Cas12a2-based RNA-detecting tool doesn't show advantage of the existing RNA-detecting tools. However, it would be interesting to develop the biotechnological tools based on the specific properties of Cas12a2, like degrading dsDNA and recognizing flexible PFS.

The reviewer correctly points out that we employ ssDNA molecular beacons similar to existing CRISPR detection technologies. While Cas12a2 is the only existing nuclease that naturally recognizes RNA targets and can cleave ssDNA at ambient temperatures (unlike Cas12g), the unique features of Cas12a2 could be better harnessed in CRISPR applications. As a proof-of-principle demonstration, we utilized plasmid DNA in place of the traditional RNA and ssDNA molecular beacons. To create a positive readout of nicking activity, we added a step involving DNA nick translation coupled to a fluorescent readout. This addition resulted in an elevated signal only in the presence of the target RNA. The resulting data (see below) were incorporated as Extended Data Fig. 19d-f.

To capture the new assay, we added the following:

Pages 12-13: “Furthermore, we devised a modified detection assay that utilizes plasmid DNA and DNA nick translation⁴⁵, introducing a distinct positive readout for CRISPR-based diagnostics (**Extended Data Fig. 19d-f**).”

4) Line 118, “Of the three collateral substrates, ssRNA and ssDNA were more efficiently cleaved than ssDNA by Cas12a2.”: Should the later ssDNA be dsDNA? If it is possible to measure the affinity of Cas12a2 to substrate (non-target RNA, non-target ssDNA and non-target dsDNA) and explain why Cas12a2 uses dsDNA to trigger SOS response on the condition that ssRNA and ssDNA were more efficiently cleaved than ssDNA.

We thank the reviewer for catching this unfortunate typo, as the second ssDNA should be dsDNA. We have updated the sentence accordingly.

We also revised the wording to highlight the different ratios of enzyme and phosphodiester backbone between dsDNA, ssDNA, and ssRNA experiments, implying how such a difference could influence the cleavage curves.

“Of the three collateral substrates, ssRNA and ssDNA appear to be more efficiently cleaved than dsDNA by Cas12a2 (**Fig. 2c and Extended Data Fig. 9b**). However, this difference could be explained by the presence of twice as many DNA strands in dsDNA substrates than ssDNA substrates for the same amount of nuclease.”

5) In Fig. 2e, the plasmid DNA was fully cleaved in 60 min, but the data in Ext. Data Fig. 9b shows that the non-target dsDNA had not been fully cleaved in 180 min. Does this mean the supercoiled dsDNA was cleaved much faster than linear dsDNA? Could the authors give some explanations for that?

We appreciate the reviewer’s careful attention to these different experiments. We believe that the discrepancy in apparent cleavage rates between the two referenced experiments can be explained by differences in experimental conditions. In particular, for the cleavage experiment with a small FAM-labeled dsDNA, the ratio of protein and nucleic acids was Cas12a2:crRNA:target RNA:non-target RNA is 1:1:1:1; in contrast, for the plasmid cleavage experiment, the reactions are 2:2:~3.6:1. Thus, in the plasmid cleavage assay, there is twice as much Cas12a2 and almost four times the amount of target RNA than in the FAM-labeled dsDNA cleavage assay, explaining the apparent increase in cleavage observed for plasmid.

6) Line 245: “Consistent with this observation, recent cryo-EM structures reveal Cas12a2 binds and cuts dsDNA with a mechanism completely distinct from all other CRISPR associated nucleases, while structure-guided mutants with impaired in vitro collateral dsDNase but not ssRNase and ssDNase activities abolished in vivo defense activity²⁰”: The data in ref. 20 is not convincing with the conclusion that “structure-guided mutants with impaired in vitro collateral dsDNase but not ssRNase and ssDNase activities abolished in vivo defense activity.” Could the authors explain why the Y465A mutant shows robust dsDNase activity but has low ability to clear plasmid? However, E1063A has completely lost dsDNA cleavage activity but has higher ability to clear plasmid than Y465A (Fig. 3h, I in ref. 20). It would be better that the authors test these mutants mentioned in ref. 20.

We thank the reviewer for pointing out this apparent discrepancy. To address this concern, we have now included additional data in the accompanying manuscript of time courses showing substantial differences in supercoiled plasmid cleavage over time (see Bravo et al. Ext. Data Fig. 8). Y465A is the most active mutant, demonstrating the ability to nick and linearize a portion of the plasmid, consistent with the dsDNA activity observed in Figure 3 of Bravo et al. However, the Y465A mutant is unable to completely destroy the plasmid DNA similar to WT. Indeed, in the presence of the Y465A mutant, more than 50% of the plasmid remains supercoiled after 60 minutes, while in the presence of WT SuCas12a2, the plasmid is completely degraded after 30 minutes. We believe these additional data provide a more detailed explanation for why the clamp

mutants are ineffective nucleases *in vivo* and provide further support for our original claim about the essential role of dsDNase activity in immune defense.

Reviewer Reports on the First Revision:

Referees' comments:

Referee #1:

The authors carefully addressed all of my and the other reviewer's comments. I do have a few last (minor) issues that need to be resolved.

The authors' response to my comment on acknowledging Abi-inducing indiscriminate nucleases in type III systems, was addressed by changing the text to read:

"Several RNA-targeting CRISPR-Cas systems (e.g., types III and VI) cause Abi phenotypes by activating indiscriminate nucleases. However, a CRISPR-mediated abortive mechanism that leverages indiscriminate DNase activity through a single-effector nuclease is yet to be observed."

These two sentences now contradict each other (as also acknowledged by the authors: Card1, a single-effector nuclease, provides an Abi phenotype by acting on DNA)". Hence, perhaps the authors meant to say that "a Class 2 CRISPR-mediated abortive....is yet to be observed" here?

The authors should be consistent in what is being depicted/quantified in Fig. 4c and Ext. Data Fig. 15. In Fig. 4c the authors label the bottom band as "tRNA", whereas in Ext. Data Fig. 15 they are labeled and quantified as "small RNAs". Judging from the authors' rebuttal, they seem to infer that the bottom band contains both tRNAs and small RNAs?

Lines 116 and 118: Change both instances of "indiscriminantly" to "indiscriminately".

Referee #2:

This manuscript has been improved with the additional experiments and revised descriptions. The authors have addressed all my concerns.

Author Rebuttals to First Revision:

Manuscript 2022-06-09197A

Rebuttal to the reviewers' comments

Referee #1:

The authors' response to my comment on acknowledging Abi-inducing indiscriminate nucleases in type III systems, was addressed by changing the text to read: "Several RNA-targeting CRISPR-Cas systems (e.g., types III and VI) cause Abi phenotypes by activating indiscriminate nucleases. However, a CRISPR-mediated abortive mechanism that leverages indiscriminate DNase activity through a single-effector nuclease is yet to be observed."

These two sentences now contradict each other (as also acknowledged by the authors: Card1, a single-effector nuclease, provides an Abi phenotype by acting on DNA)". Hence, perhaps the authors meant to say that "a Class 2 CRISPR-mediated abortive...is yet to be observed" here?

We thank the reviewer for their additional efforts to improve our work. Regarding the two sentences, our intention was for "single-effector nuclease" to be understood as a RNA-guided Cas effector used as the primary means of immune defense, whereas Card1 is an accessory nuclease activated by the triggered Type III multi-effector complex. To better convey this distinction, we rephrased this part of the abstract to read the following:

"Several RNA-targeting CRISPR-Cas systems (i.e., types III and VI) cause Abi phenotypes by activating indiscriminate nucleases³⁻⁵. However, a CRISPR-mediated abortive mechanism that leverages indiscriminate DNase activity of an RNA-guided single-effector nuclease has yet to be observed. Here we report that RNA targeting by the type V single-effector nuclease called Cas12a2 drives abortive infection through non-specific cleavage of double-stranded (ds)DNA."

The authors should be consistent in what is being depicted/quantified in Fig. 4c and Ext. Data Fig. 15. In Fig. 4c the authors label the bottom band as "tRNA", whereas in Ext. Data Fig. 15 they are labeled and quantified as "small RNAs". Judging from the authors' rebuttal, they seem to infer that the bottom band contains both tRNAs and small RNAs?

We thank the reviewer for catching this discrepancy. In response to this comment, we updated Fig. 4c and ED Fig. 7 and their corresponding legends to only read "sRNA pool" and changed the text in the Results section as follows to improve clarity:

"While Cas13a significantly depleted both rRNAs and the small RNA pool that includes tRNAs, Cas12a2 significantly depleted only the small RNA pool."

Lines 116 and 118: Change both instances of "indiscriminantly" to "indiscriminately".

We thank the reviewer for catching the misspelling, which we have corrected.

Referee #2:

This manuscript has been improved with the additional experiments and revised descriptions. The authors have addressed all my concerns.

We thank the reviewer for their supportive comments